# From ERA-Interim to ERA5: considerable impact of ECMWF's next-generation reanalysis on Lagrangian transport simulations

Lars Hoffmann[1], Gebhard Günther[2], Dan Li[2,3], Olaf Stein[1], Xue Wu[1,3], Sabine Griessbach[1], Yi Heng[4], Paul Konopka[2], Rolf Müller[2], Bärbel Vogel[2], and Jonathon S. Wright[5]

[1]Jülich Supercomputing Centre, Forschungszentrum Jülich, Jülich, Germany
[2]Institut für Energie- und Klimaforschung (IEK-7), Forschungszentrum Jülich, Jülich, Germany
[3]Key Laboratory of Middle Atmosphere and Global Environment Observation, Institute of Atmospheric Physics, Chinese Academy of Sciences, Beijing, China
[4]School of Data and Computer Science, Sun Yat-sen University, Guangzhou, China
[5]Department of Earth System Science, Tsinghua University, Beijing, China

**Correspondence:** Lars Hoffmann (l.hoffmann@fz-juelich.de)

**Abstract.** The European Centre for Medium-Range Weather Forecasts' (ECMWF's) next-generation reanalysis ERA5 provides many improvements, but it also confronts the community with a 'big data' challenge. Data storage requirements for ERA5 increase by a factor of ∼80 compared with the ERA-Interim reanalysis, introduced a decade ago. Considering the significant increase in resources required for working with the new ERA5 data set, it is important to assess its impact on Lagrangian transport simulations. To quantify the differences between transport simulations using ERA5 and ERA-Interim data, we analyzed comprehensive global sets of 10-day forward trajectories for the free troposphere and the stratosphere for the year 2017. The new ERA5 data have considerable impact on the simulations. Spatial transport deviations between ERA5 and ERA-Interim trajectories are up to an order of magnitude larger than those caused by parameterized diffusion and subgrid-scale wind fluctuations after 1 day and still up to a factor of 2 – 3 larger after 10 days. Depending on the height range, the spatial differences between the trajectories map into deviations as large as 3 K in temperature, 30% in specific humidity, 1.8% in potential temperature, and 50% in potential vorticity after 1 day. Part of the differences between ERA5 and ERA-Interim is attributed to better spatial and temporal resolution of the ERA5 reanalysis, allowing for a better representation of convective updrafts, gravity waves, tropical cyclones, and other meso- to synoptic scale features of the atmosphere. Another important finding is that ERA5 trajectories exhibit significantly improved conservation of potential temperature in the stratosphere, pointing to an improved consistency of ECMWF's forecast model and observations that leads to smaller data assimilation increments. We conducted a number of downsampling experiments with the ERA5 data, in which we reduced the numbers of meteorological time steps, vertical levels, and horizontal grid points. Significant differences remain present in the transport simulations, if we downsample the ERA5 data to a resolution similar to ERA-Interim. This points to substantial changes of the forecast model, observations, and assimilation system of ERA5 in addition to improved resolution. A comparison of two Lagrangian trajectory models allowed us to assess the readiness of the codes and workflows to handle the comprehensive ERA5 data and to demonstrate the consistency of the simulation results. Our results will help to guide future Lagrangian transport studies attempting to

navigate the increased computational complexity and leverage the considerable benefits and improvements of ECMWF's new ERA5 data set.

## 1 Introduction

Lagrangian transport models are indispensable tools for studying atmospheric transport processes (e. g., Djurić, 1961; Hsu, 1980; Kida, 1983; Thomson, 1987; Wernli and Davies, 1997; Draxler and Hess, 1998; McKenna et al., 2002a, b; Legras et al., 2003; Lin et al., 2003; Stohl et al., 2005; Jones et al., 2007; Lin et al., 2012; Bowman et al., 2013, and references therein). Lagrangian transport models simulate the dispersion of trace gases or aerosols by means of trajectory calculations for a number of infinitesimally small air parcels or 'particles' following the fluid flow. A major advantage is that the spatial resolution of Lagrangian transport simulations is not limited to a regular grid. The approach can avoid the numerical diffusion of passive tracers that is always present to some degree in Eulerian models. The method is therefore well capable of representing small-scale features such as filaments of tracers associated with long-range transport. Because of their distinct advantages, Lagrangian transport models have found a variety of operational and research applications. For example, the authors of this study have recently applied Lagrangian transport models to study transport pathways associated with the Asian summer monsoon (e. g., Konopka et al., 2010; Wright et al., 2011; Ploeger et al., 2013; Vogel et al., 2016; Li et al., 2017) and the dispersion of ash and sulfur dioxide plumes from volcanic eruptions (Heng et al., 2016; Hoffmann et al., 2016; Wu et al., 2017, 2018).

Lagrangian transport simulations are typically driven by external data from meteorological reanalyses or operational forecasts. A comprehensive overview of state-of-the-art American, European, and Japanese reanalyses was recently presented by Fujiwara et al. (2017). Meteorological data sets provided by the European Centre for Medium-Range Weather Forecasts (ECMWF) are among those data frequently used for Lagrangian transport simulations. In 2006, the ECMWF implemented the ERA-Interim reanalysis (Dee et al., 2011), which has since been successfully applied in thousands of research applications. About a decade later, ECMWF implemented the successor of ERA-Interim, its 5th-generation reanalysis, referred to as ERA5 (Hersbach and Dee, 2016). This new reanalysis comes with many improvements compared with ERA-Interim, most notably better spatial and temporal resolution (see Table 1), but also other aspects, such as better representation of geophysical processes in the forecast model and more extensive observational inputs to the data assimilation system.

However, the new ERA5 products pose significant technical challenges for Lagrangian transport model simulations. The application of ERA5 at its full spatiotemporal resolution comes along with a substantial increase in computing resources and storage requirements. For example, the computational time and main memory requirements increase by a factor of ∼10 and the total disk space required for input data increases by a factor of ∼80 for a typical simulation conducted for this study, as we progress from ERA-Interim to ERA5 (Table 1). The increase in disk space size is mostly due to the better spatiotemporal resolution of the ERA5 data, i. e., a factor of 6 in the number of synoptic time steps, a factor of 2.2 in the number of vertical levels, and a factor of $2.5 \times 2.5$ in the number of horizontal grid points. While this might be acceptable for trajectory studies covering short time periods, the capabilities to conduct comprehensive global simulations (e. g., Vogel et al., 2016), long-term simulations for climate studies (e. g., Pommrich et al., 2014; Tao et al., 2015; Konopka et al., 2016), or ensemble runs for

inverse modeling studies (e. g., Heng et al., 2016) are hampered by these demands. In this paper, we describe some of the changes of the models and workflows that are necessary to cope with the increase in computational requirements, in particular the increase in storage requirements. The particular benefits that come along with using the next-generation ECMWF reanalysis have been carefully evaluated.

The main aim of this study is to quantify the impact of the new ERA5 data on Lagrangian transport simulations. Considering the significant computing resources required to conduct simulations with ERA5 data, our study is limited to comparisons for a single year. More specifically, we quantified the differences between ERA5 and ERA-Interim driven simulations for different height ranges in the free troposphere and stratosphere for a set of 24 simulations for the year 2017, each covering up to 10 days of simulation time. The statistical analysis covers spatial differences between the trajectories as well as differences in meteorological variables and dynamical tracers such as temperature, specific humidity, potential temperature, and potential vorticity along the trajectories. We provide a number of examples illustrating the differences between ERA5 and ERA-Interim simulations in practice. Downsampling experiments were conducted, as downsampling can potentially help to mitigate some of the problems associated with increased computational overhead of the ERA5 simulations and to distinguish between the impact of improved resolution and other changes in the reanalysis system. We evaluated the readiness of two Lagrangian trajectory models, the Chemical Lagrangian Model of the Stratosphere (CLaMS) (McKenna et al., 2002a, b) and Massive-Parallel Trajectory Calculations (MPTRAC) (Hoffmann et al., 2016), to operate with ERA5 data and compared the simulation results. Obviously, this study can cover only some of the potential applications of Lagrangian transport models, but its outcome may help to guide future studies regarding the increased computational resources and possible benefits and improvements related to the new ERA5 data.

In Sect. 2 we provide descriptions of the ERA5 and ERA-Interim reanalyses, the meteorological conditions during the year 2017, the CLaMS and MPTRAC models, the simulation setups for the numerical experiments, and the statistical measures used to evaluate the transport simulations. Section 3 presents the results of the study, covering analyses of the impacts of parameterized diffusion and subgrid-scale wind fluctuations, transport deviations between ERA5 and ERA-Interim, dynamical tracer conservation, downsampling experiments, and a comparison of CLaMS and MPTRAC model simulations. A brief discussion and conclusions are given in Sect. 4.

## 2 Data and methods

### 2.1 Meteorological data

#### 2.1.1 The ERA-Interim and ERA5 reanalyses

The ERA-Interim reanalysis (Dee et al., 2011) is a global atmospheric reanalysis covering the time period from 1979 to present, with continuous updates in near real time up to the present day. The reanalysis is produced using ECMWF's Integrated Forecast System (IFS) Cycle 31r2, which was released in 2006. The horizontal resolution of the data set is $\sim$79 km ($T_L$255 spectral grid) on 60 model levels from the surface up to 0.1 hPa (about 65 km of altitude). For this study, we retrieved the ERA-Interim data

at $0.75° \times 0.75°$ horizontal sampling and on all model levels from ECMWF. The system applies 4-dimensional variational analysis (4D-Var) with a 12-hour analysis window. The ERA-Interim analyses are provided for 0, 6, 12, and 18 UTC. Global atmospheric budgets of mass, moisture, energy, and angular momentum were studied in detail by Berrisford et al. (2011), reporting significant improvements compared to the earlier ERA-40 reanalysis (Uppala et al., 2005).

The next-generation ERA5 reanalysis will eventually cover the time period from January 1950 to present. At the time of this writing, a first segment of data from 2000 to near present has been made available to the public. The ERA5 reanalysis is produced using the IFS Cycle 41r2 with 4D-Var data assimilation, as released in 2016. Part of ERA5 is a high resolution realization atmospheric data set with a horizontal resolution of $\sim$31 km ($T_L$639 spectral grid). The data are provided on 137 hybrid sigma/pressure levels in the vertical, with the top level located at 0.01 hPa (about 80 km of altitude). We retrieved the

data at $0.3° \times 0.3°$ horizontal sampling and on all model levels from ECMWF. The system provides hourly estimates of a comprehensive number of atmospheric, terrestrial, and oceanic climate variables.

    ERA5 will eventually replace the ERA-Interim reanalysis, with the production period of ERA-Interim potentially ending as early as 2018 (Hersbach and Dee, 2016). According to the ECMWF, ERA5 improves upon ERA-Interim in various aspects. One of the major improvements of ERA5 is the much higher spatial and temporal resolution. Figure 1 illustrates the improved

vertical coverage and sampling of ERA5 compared with ERA-Interim. Furthermore, the representation of tropospheric processes appears to be significantly improved in ERA5, including better representation of tropical cyclones, better global balance of precipitation and evaporation, better precipitation over land in the deep tropics, better soil moisture, and more consistent sea surface temperatures and sea ice (Hennermann and Berrisford, 2018). In contrast to ERA-Interim, ERA5 includes a lower-resolution 10-member ensemble of data assimilations that provides additional information on uncertainties in the reanalysis

and their changes over space and time. More detailed descriptions of the ECMWF reanalyses and their differences can be found in Dee et al. (2011), Hersbach and Dee (2016), and the upcoming final report of the Stratosphere-Troposphere Processes and their Role in Climate (SPARC) Reanalysis Intercomparison Project (S-RIP) (Fujiwara et al., 2017).

### 2.1.2   Meteorological conditions during the year 2017

In this section we briefly describe some of the meteorological events and conditions that occurred in the free troposphere and

stratosphere during the year 2017 based on reports by Blunden and Arndt (2018), Krummel et al. (2018), and WMO (2018) as well as public information provided by the National Aeronautics and Space Administration (https://ozonewatch.gsfc.nasa.gov, last access: 14 November 2018). Illustrative examples of ERA5 and ERA-Interim data for the year 2017 are shown in Figs. 2 to 4. Figure 2 shows ERA5 and ERA-Interim maps of horizontal wind speed, vertical velocity, and potential vorticity at 500 hPa (about 5 km of altitude) over North America and over the North Atlantic on 8 September 2017, a day with exceptional

hurricane activity over the North Atlantic. Figures 3 and 4 show zonal mean temperatures, water vapor volume mixing ratios, and zonal winds for ERA5 and their differences with respect to ERA-Interim in January and July 2017, respectively.

    The year 2017 was one of the three warmest years in the troposphere on record, slightly below the levels of 2015 and 2016, and it was the warmest year that was not influenced by an El Niño event. A neutral phase of the El Niño Southern Oscillation prevailed for most of 2017, evolving into a weak La Niña by November. Over the Arctic, sea-ice extent was well below

average throughout 2017, with record-low levels during the first four months of the year. In 2017, 84 tropical cyclones were observed globally, very close to the long-term average. However, the hurricane season in the North Atlantic was exceptional. In 2017, the North Atlantic had 17 named storms, and the value of accumulated cyclone energy ranked seventh on record, including a record-high monthly value for September. Three exceptionally destructive hurricanes occurred in rapid succession

over the North Atlantic in late August and September, namely Harvey (category 4, 17 August – 2 September), Irma (category 5, 6 – 12 September), and Maria (category 5, 16 September – 2 October). Figure 2 illustrates that the representation of tropical storms is significantly improved in ERA5 relative to ERA-Interim. In particular, ERA5 shows stronger and more realistic horizontal wind speeds, vertical velocities, and potential vorticities. This is promising, because tropical storm intensities are often underrepresented in earlier reanalyses (Hodges et al., 2017). Furthermore, Fig. 2 also suggests that ERA5 better resolves

individual convective updrafts over land and near the Intertropical Convergence Zone (ITCZ) as well as other small-scale features, such as gravity waves.

Considering the stratosphere, the phase of the quasi-biennial oscillation (QBO) was mainly westerly at both 30 and 50 hPa until June 2017, at which point the wind anomalies at 30 hPa reversed. At Northern Hemisphere high latitudes, there was a brief major mid-winter warming in early February and another warming in early March. At these times, the polar vortex in

the Northern Hemisphere was distorted and displaced from the pole. In November, the polar vortex was of average size and strength, but became distorted and more disturbed than the climatological mean state in December. In the Southern Hemisphere, the polar vortex became unstable and elliptical in the third week of September, with a sudden decrease of polar wind speed, with temperatures within the polar cap (60 – 90°S) attaining the maximum value on record from 1979 to 2017. The 2017 Antarctic ozone hole was slightly smaller than the long-term mean of 1979 to 2017, and the warming in September resulted in a rapid

decrease of its size. The comparison of zonal mean zonal winds and temperatures in Figs. 3 and 4 suggests that large-scale features are represented equally well in ERA5 and ERA-Interim. Notable differences appear only in the upper stratosphere, where ERA-Interim has substantially lower vertical resolution than ERA5. A different representation of gravity waves and the QBO in ERA5 (Orr et al., 2010) may explain the differences seen in the tropical zonal winds. The temperature biases between ERA5 and ERA-Interim in the upper stratosphere are possibly related to different treatment of satellite observations in the data

assimilation schemes. The comparison of water vapor volume mixing ratios in Figs. 3 and 4 shows a substantial high bias of up to 25% for ERA-Interim compared to ERA5 in the lowermost stratosphere at mid and high latitudes. This may indicate that the new version of the ECMWF reanalysis has less leakage of water vapor into the extratropical lowermost stratosphere, which reduces known moist biases of earlier ECMWF data sets in this region (Dyroff et al., 2015). Also, the Southern Hemisphere lower polar vortex in ERA5 in July 2017 was notably dryer than the one in ERA-Interim.

## 2.2 Lagrangian transport models

We conducted the Lagrangian transport simulations for this study with two models. MPTRAC (Hoffmann et al., 2016) has been developed recently to support analyses of atmospheric transport processes in the free troposphere and stratosphere. MPTRAC features a modular structure for different geophysical processes. Most importantly, the advection module of MPTRAC solves the trajectory equation for atmospheric air parcels based on given wind fields from ERA5, ERA-Interim, or other meteorolog-

ical data sets. Kinematic trajectories are calculated using pressure as the vertical coordinate. Another module is available to simulate diffusion and subgrid-scale wind fluctuations by adding stochastic perturbations to the trajectories, following the approach of Stohl et al. (2005). Additional modules can simulate sedimentation (i. e., gravitational settling) or the decay of mass assigned to the air parcels. MPTRAC is particularly suited for large-scale simulations on supercomputers due to its Message
Passing Interface (MPI)/Open Multi-Processing (OpenMP) hybrid parallelization. Among the first applications, MPTRAC was used to perform Lagrangian transport simulations of the dispersion of volcanic plumes and to estimate sulfur dioxide emission rates for these events (Heng et al., 2016; Hoffmann et al., 2016; Wu et al., 2017, 2018). Hoffmann et al. (2017) presented an intercomparison of meteorological analyses and an evaluation of MPTRAC trajectory calculations with superpressure balloon observations for the Antarctic lower stratosphere. Rößler et al. (2018) evaluated trajectory errors of different numerical inte-
gration schemes diagnosed with the MPTRAC advection module driven by high-resolution ECMWF operational analyses and forecasts.

In this study, we also applied the Chemical Lagrangian Model of the Stratosphere (CLaMS) trajectory module (Sutton et al., 1994; McKenna et al., 2002b) to calculate kinematic forward trajectories. CLaMS performs the fully Lagrangian, non diffusive, 3-dimensional advection of an ensemble of air parcels (Konopka et al., 2004; Pommrich et al., 2014). Combined with additional
15 modules to represent mixing of air masses, CLaMS is well suited for reproducing atmospheric transport barriers, such as the edge of the polar vortex (Konopka et al., 2004, 2005; Hoppe et al., 2014) and the Asian summer monsoon anticyclone (Konopka et al., 2010; Ploeger et al., 2013, 2015; Vogel et al., 2015, 2016). The trajectories of air parcels are calculated using the classical 4th-order Runge-Kutta method with a $600\,\mathrm{s}$ time step for simulations based on ERA-Interim and $240\,\mathrm{s}$ for simulations based on ERA5. The same time steps were used for MPTRAC, applying the midpoint method to solve the trajectory equation.
Sensitivity tests showed that the time steps are small enough so that truncation errors do not contribute significantly to the simulation results. Like MPTRAC, the CLaMS trajectory module employs pressure (interpolated from the ECMWF hybrid vertical coordinate) as the vertical coordinate along with vertical velocity, $\omega = dp/dt$, as provided by ECMWF to calculate kinematic trajectories. Alternatively, the CLaMS trajectory module can be used to calculate diabatic trajectories. Although diabatic trajectories have known advantages for the upper troposphere and stratosphere (e. g., Ploeger et al., 2010, 2011; Tissier
and Legras, 2016), they are rarely used for the lower and middle troposphere. A comparison of diabatic and kinematic trajectory calculations is beyond the scope of our present work, which focuses exclusively on kinematic forward trajectories.

## 2.3 Evaluation of transport simulations

### 2.3.1 Simulation setup and overview of numerical experiments

In order to evaluate the impact of different meteorological data sets or different model configurations on the Lagrangian
transport simulations, we conducted various experiments based on a set of 24 simulations, starting on the 1st and 15th of each month of the year 2017. In each simulation we calculated 10-day forward trajectories for $10^6$ particles. The trajectory seeds were distributed globally, with a density based on cosine-weighting of latitude to achieve quasi-equidistant horizontal sampling. The initial vertical distribution of the seeds was uniform within the log-pressure altitude range of $2-48\,\mathrm{km}$. We

did not perform any simulations for particles launched below 2 km, because both CLaMS and MPTRAC lack sophisticated parameterizations of diffusion within the planetary boundary layer. We restricted the initial upper altitude to 48 km, because tests showed large discrepancies between ERA5 and ERA-Interim above the stratopause, likely due to the low number of levels and strong model constraints of ERA-Interim in the lower mesosphere. We sampled temperature, specific humidity, potential temperature, and potential vorticity along the trajectories. The simulation output was saved every 6 hours.

Following the approach of Rößler et al. (2018), we evaluated the simulation results separately in different height ranges and latitude bands. Considering that the trajectory errors depend on the height level within the atmosphere, we split the full log-pressure altitude range of 2 – 48 km into four layers. Roughly, these layers cover the free troposphere (2 – 8 km), the upper troposphere and lower stratosphere (UT/LS, 8 – 16 km), the lower and middle stratosphere (16 – 32 km), and the middle and upper stratosphere (32 – 48 km). For the UT/LS region, this definition is particularly limited, as the UT/LS region may cover height ranges from roughly 5 to 22 km in reality (Eyring et al., 2010). Rößler et al. (2018) found that trajectory errors within different height layers also vary with latitude and season. We therefore evaluated the simulation results not only globally, but also in three latitude bands, covering the Northern Hemisphere extratropics (20°N – 90°N), the tropics (20°S – 20°N), and the Southern Hemisphere extratropics (20°S – 90°S). We did not separate between mid and high latitudes, because trajectories frequently meander between these latitude bands due to the jet streams, making it difficult to attribute the trajectory errors to different latitude bands. Here, the binning of the particles into the different height ranges and latitude bands was performed at each time step according to their actual positions along the trajectories.

### 2.3.2 Statistical analysis of transport deviations

Various statistical quantities have been proposed to measure the differences between sets of test and reference trajectories. Spatial differences of trajectories are commonly measured in terms of absolute horizontal and vertical transport deviations (AHTD and AVTD, Kuo et al., 1985; Rolph and Draxler, 1990; Stohl, 1998). Considering two sets of $N$ trajectories each, with particle positions $[x_i(t_n), y_i(t_n), z_i(t_n)]$ and $[X_i(t_n), Y_i(t_n), Z_i(t_n)]$, the AHTD and AVTD at a time step $t_n$ are

$$\text{AHTD}(t_n) \quad = \quad \frac{1}{N} \sum_{i=1}^{N} \sqrt{[x_i(t_n) - X_i(t_n)]^2 + [y_i(t_n) - Y_i(t_n)]^2}, \tag{1}$$

$$\text{AVTD}(t_n) \quad = \quad \frac{1}{N} \sum_{i=1}^{N} |z_i(t_n) - Z_i(t_n)|. \tag{2}$$

Here, the horizontal distances are calculated by converting the geographic longitudes and latitudes of the particles to Cartesian coordinates, followed by calculation of the Euclidean distance of the Cartesian coordinates. Euclidean distances approximate great circle distances with good accuracy ($\geq 97\%$ up to 5000 km of distance). Vertical distances are calculated based on conversion of particle pressure to log-pressure altitude using the barometric formula. Note that all altitudes reported in this paper are log-pressure altitudes, calculated from the barometric formula with a constant surface pressure of 1013.25 hPa and a scale height of 7 km. The Lagrangian models themselves operate on pressure levels.

Considering the mean horizontal and vertical path lengths of individual trajectories ($L_{h,i}$ and $L_{v,i}$) of the test and reference data set integrated over the time steps $t_1, \ldots, t_n$,

$$L_{h,i}(t_n) = \frac{1}{2} \sum_{j=2}^{n} \left\{ \sqrt{[x_i(t_j) - x_i(t_{j-1})]^2 + [y_i(t_j) - y_i(t_{j-1})]^2} \right.$$
$$\left. + \sqrt{[X_i(t_j) - X_i(t_{j-1})]^2 + [X_i(t_j) - X_i(t_{j-1})]^2} \right\}, \tag{3}$$

$$L_{v,i}(t_n) = \frac{1}{2} \sum_{j=2}^{n} \left\{ \left| z_i(t_j) - z_i(t_{j-1}) \right| + \left| Z_i(t_j) - Z_i(t_{j-1}) \right| \right\}, \tag{4}$$

the corresponding relative horizontal and vertical transport deviations (RHTD and RVTD) are

$$\text{RHTD}(t_n) = \frac{1}{N} \sum_{i=1}^{N} \frac{\sqrt{[x_i(t_n) - X_i(t_n)]^2 + [y_i(t_n) - Y_i(t_n)]^2}}{L_{h,i}(t_n)}. \tag{5}$$

$$\text{RVTD}(t_n) = \frac{1}{N} \sum_{i=1}^{N} \frac{|z_i(t_n) - Z_i(t_n)|}{L_{v,i}(t_n)}. \tag{6}$$

Stohl (1998) pointed out that there are some ambiguities in how RHTDs and RVTDs are defined in the literature. Careful attention should be paid to the definitions of the RHTD and RVTD, when the results of different studies are compared to each other. We point out that the temporal sampling between the time steps $t_j$ also matters, as it determines how much of the horizontal meandering and vertical oscillations of the trajectories are captured. Here, the sampling interval of the trajectory output was set to 6 h.

In addition to the transport deviations, we evaluated the deviations of meteorological variables and dynamical tracers along the trajectories, including temperature, specific humidity, potential temperature, and potential vorticity. To quantify the differences of the variables $q_i$ and $Q_i$ along the test and reference trajectories, respectively, we calculated either the mean absolute deviation (MAD) or the mean relative deviation (MRD),

$$\text{MAD}(t_n) = \frac{1}{N} \sum_{i=1}^{N} |q_i(t_n) - Q_i(t_n)|, \tag{7}$$

$$\text{MRD}(t_n) = \frac{2}{N} \sum_{i=1}^{N} \frac{|q_i(t_n) - Q_i(t_n)|}{|q_i(t_n)| + |Q_i(t_n)|}. \tag{8}$$

Here, we chose MADs rather than standard deviations for the statistical analysis to achieve consistency with the definitions of the transport deviations (AHTDs and AVTDs). Also, MADs are more robust than standard deviations against outliers. For a more detailed discussion of the advantages and disadvantages of using MADs versus standard deviations see Willmott and Matsuura (2005) and Chai and Draxler (2014).

Next to the MADs and MRDs, we also evaluated the absolute bias (BA) and relative bias (BR) of the of meteorological variables and dynamical tracers along the trajectories,

$$\text{BA}(t_n) = \frac{1}{N} \sum_{i=1}^{N} [q_i(t_n) - Q_i(t_n)], \tag{9}$$

$$\text{BR}(t_n) = \frac{2}{N} \sum_{i=1}^{N} \frac{q_i(t_n) - Q_i(t_n)}{|q_i(t_n)| + |Q_i(t_n)|}. \tag{10}$$

The absolute and relative bias indicate whether systematic differences are present between the means of the distributions, whereas MADs and MRDs are measures of variability of the differences. Note that in our definitions the BR and MRD are calculated by dividing through the mean of the magnitudes of $q_i$ and $Q_i$ rather than the magnitude of the mean. This specific approach helps to solve problems with outliers when calculating the BRs or MRDs for potential vorticity in the tropics, where absolute values are small and potential vorticity changes sign.

Considering that some of the meteorological variables in this study are dynamical tracers that can be conserved along the trajectories, we also evaluated the relative tracer conservation errors (RTCE) of individual trajectory sets,

$$\text{RTCE}(t_n) = \frac{2}{N} \sum_{i=1}^{N} \frac{|q_i(t_n) - q_i(t_1)|}{|q_i(t_n)| + |q_i(t_1)|}. \tag{11}$$

Note that in reality part of the RTCE is due to non-conservation, e. g., due to diabatic heating or dissipation. This analysis follows the approach of Stohl and Seibert (1998), but we restricted the calculation of the RTCE to the change of the tracer quantities between the time steps $t_1$ and $t_n$ of the trajectories rather than integrating over all possible combinations of $t_i$ and $t_j$ along the trajectories, because of the large number of particles that was considered in this study.

## 3 Results

### 3.1 Impact of diffusion on ERA5 trajectories

In this section, we analyze the impact of the diffusion and subgrid-scale wind fluctuation parameterizations in MPTRAC on the Lagrangian transport simulations. Quantifying the impact of diffusion and subgrid-scale wind fluctuations is particularly helpful, because it provides us with a reference for assessing the impact of other effects on the Lagrangian transport simulations. For example, comparing the deviations between ERA5 and ERA-Interim simulations to the deviations due to diffusion and subgrid-scale wind fluctuations allows us to assess, whether the differences found between the meteorological data sets can be considered significant or not. This approach is similar to the concept of significance rating by means of the 'meteorological complexity factor' of Kahl (1996). Unfortunately, a difficulty arises from the fact that the strength of dispersion modeled with the approach of Stohl et al. (2005) depends on the particular meteorological data set (Hoffmann et al., 2017). Tests showed that the spread of particles in terms of AHTDs and AVTDs with respect to trajectories calculated without diffusion and subgrid-scale wind fluctuations modeled with ERA5 is about a factor of 2 lower compared with ERA-Interim. However, ERA5 provides higher spatiotemporal resolution and potentially bears lower uncertainty on the subgrid scales. Hence, we selected diffusion and subgrid-scale wind fluctuation simulations based on ERA5 as a reference for further comparisons. ERA5 data provide a stricter measure of significance in our assessment, as trajectories based on ERA5 have a lower spread than those based on ERA-Interim.

Figure 5 provides an illustrative example of the impacts of parameterized diffusion and subgrid-scale wind fluctuations on the Lagrangian transport simulations. The figure shows ERA5 10-day forward trajectories with and without diffusion and subgrid-scale wind fluctuations for a single seed in the mid-latitude lower stratosphere in Northern Hemisphere winter. A more detailed analysis showed that the dispersion of the ERA5 trajectory set seen in this particular example is mostly due

to a combination of vertical displacements due to the use of a constant vertical diffusivity $D_z = 0.1\,\mathrm{m}^2/\mathrm{s}$ in the stratosphere (Legras et al., 2003; Stohl et al., 2005) and vertical shear of the resolved horizontal winds. Note that the resulting horizontal and vertical distributions of the particle positions became non-Gaussian. For comparison, the ERA-Interim trajectory without diffusion and subgrid-scale wind fluctuations is also shown. In this example, we found particularly good agreement between the positions of the ERA5 and ERA-Interim trajectories without diffusion and subgrid-scale wind fluctuations at all times (AHTD $\leq 250\,\mathrm{km}$ and AVTD $\leq 600\,\mathrm{m}$, Figs. 5a and 5b). The ERA5 trajectory set with diffusion and subgrid-scale wind fluctuations shows a large spread that typically exceeds the differences between the ERA5 and ERA-Interim trajectories without diffusion and subgrid-scale wind fluctuations. The spatial differences between the reference trajectories without diffusion and subgrid-scale wind fluctuations can therefore be attributed to the meteorological complexity of the situation rather than to significant differences between the ERA5 and ERA-Interim data set in this case.

Figure 5 also shows differences of meteorological variables sampled along the trajectories. Starting from an initial temperature bias of 0.9 K between ERA-Interim and ERA5, temperature deviations mostly remain below 2.5 K along the trajectories (Fig. 5c). The ERA5 trajectory reveals larger temperature variability than the ERA-Interim trajectory, owing to the better spatiotemporal resolution of the ERA5 data possibly providing an improved representation of small-scale features. Significant differences are observed for water vapor volume mixing ratios, which remain nearly constant at 4.6 ppmv for ERA5, but vary between $4.3 - 4.55$ ppmv for ERA-Interim (Fig. 5d). The differences between ERA5 and ERA-Interim water vapor volume mixing ratios exceed the spread of the ERA5 trajectory set. Considering that this is a stratospheric trajectory, the nearly constant water volume mixing ratio for ERA5 looks more realistic. Increased water vapor volume mixing ratios in ERA5 are promising, as ERA-Interim was previously found to have a cold and dry bias in the UT/LS region (Schoeberl et al., 2012). Similar to the characteristics of water vapor, potential temperature along the trajectory remains nearly constant at 485 K for ERA5 compared with variations between $460 - 500$ K for ERA-Interim (Fig. 5e). Again, the simulation result for ERA5 looks more realistic, considering that potential temperature typically is an excellent dynamical tracer in the stratosphere. Potential vorticity shows larger variations than potential temperature in this particular example, remaining mostly in a range of $20 - 30$ PVU for both ERA5 and ERA-Interim (Fig. 5f). As potential temperature is nearly constant in this case, the variability in potential vorticity is due to variability in relative vorticity as calculated from the horizontal winds and variability in absolute vorticity due to the particles being dispersed to different latitudes.

The transport deviations of individual trajectories depend strongly on the meteorological situation. In order to obtain statistically meaningful results, we averaged over large numbers of trajectories; i.e., $10^6$ particles distributed globally in the free troposphere and stratosphere. As an example, Fig. 6 shows the transport deviations due to diffusion and subgrid-scale wind fluctuations in different height ranges for 10-day forward trajectories started on 1 July 2017. The AHTDs grow steadily over time, indicating that this behavior is statistically robust, with maximum values of 1400 km for the troposphere and UT/LS region $(2 - 16\,\mathrm{km})$, 1100 km for the middle and upper stratosphere $(32 - 48\,\mathrm{km})$, and 500 km for the lower and middle stratosphere $(16 - 32\,\mathrm{km})$ after 10 days (Fig. 6a). Except for an initial phase of about $0.5 - 1$ day, where individual horizontal trajectory lengths are rather short, the RHTDs also grow steadily over time. After about 3 to 4 days, the RHTDs consistently decrease with increasing altitude, showing the reduced impacts of diffusion and subgrid-scale wind fluctuations with height. RHTD

maxima after 10 days decrease from 14% in the troposphere to 4% in the upper stratosphere (Fig. 6b). AVTDs also grow steadily over time, but initially exhibit a distinct scaling behavior of AVTD $\propto \sqrt{t}$ in the stratosphere (Fig. 6c). We attribute this to the approach of Stohl et al. (2005) used to simulate diffusion in MPTRAC, as this approach applies a constant vertical diffusivity of $D_z = 0.1\,\mathrm{m^2\,s^{-1}}$ in the stratosphere (following Legras et al., 2003). At later times, an exponential regime charac-
teristic of chaotic dispersion and a linear regime due to large eddy dispersion are observed. As vertical trajectory lengths are rather short initially, RVTDs tend to be largest in the beginning (up to 74% after 6 h in the lower and middle stratosphere), but converge towards much smaller values of $6-10\%$ after 10 days at all heights (Fig. 6d).

Figure 7 illustrates seasonal and latitudinal variations of the transport deviations due to parameterized diffusion and subgrid-scale wind fluctuations. It shows AHTDs and RHTDs after 10 days for each of the 24 simulations during the year 2017
for the Northern Hemisphere and Southern Hemisphere extratropics. In the AHTDs we found a strong annual cycle with wintertime maxima in the middle and upper stratosphere and peak-to-peak variations in the range of $200-2200\,\mathrm{km}$ (Figs. 7a and 7c). This seasonal cycle is plausible, considering that the wintertime stratosphere is generally more disturbed and affected by planetary wave activity in the vicinity of the polar vortex relative to the summertime stratosphere. Weaker annual cycles are present in the lower and middle stratosphere (wintertime maxima, AHTDs of $300-800\,\mathrm{km}$ in both hemispheres) and the
UT/LS region (summertime maxima, AHTDs of $800-1300\,\mathrm{km}$ at $90°\mathrm{S}-20°\mathrm{S}$ and $1100-1600\,\mathrm{km}$ at $20°\mathrm{N}-90°\mathrm{N}$). In the extratropical troposphere the AHTDs due to diffusion and subgrid-scale wind fluctuations are generally large ($1500-1900\,\mathrm{km}$ in both hemispheres), but no annual cycle was evident. Annual cycles are also present in the RHTDs (Figs. 7b and 7d), but the peak-to-peak variations are different compared with the AHTDs. We found that the annual cycles in the RHTDs are more pronounced in the troposphere (RHTDs of $10-16\%$) and UT/LS region ($5-12\%$) and less pronounced in the lower and middle
stratosphere ($4-7\%$) and the middle and upper stratosphere ($2-9\%$). A direct influence of specific meteorological conditions can be seen in the strong variations of the AHTDs in the Southern Hemisphere extratropical stratosphere from August to October 2017 (Fig. 7c), which coincides with a strong sudden stratospheric warming and associated weakening of the zonal winds in September 2017.

## 3.2  Spatial differences of ERA5 and ERA-Interim trajectories

Figure 8 provides a statistical summary of the transport deviations between the ERA-5 and ERA-Interim trajectories for the year 2017, showing the existence of significant differences between these two data sets. Figure 8 shows the median as well as the peak-to-peak range (minimum to maximum) of individual transport deviations during the course of the year. As mentioned earlier, transport deviations are shown separately for four height ranges, as well as globally, for the Northern Hemisphere extratropics, the Southern Hemisphere extratropics, and the tropics. Large peak-to-peak ranges are associated with the presence
of seasonal cycles in the data (see Sect. 3.1 and Fig. 7). Transport deviations due to parameterized diffusion and subgrid-scale wind fluctuations are shown for reference in Fig. 8. We decided to analyze the transport deviations after both 1 and 10 days. The transport deviations after 1 day are most indicative of the specific differences between ERA5 and ERA-Interim in this case. Transport deviations after 10 days can be thought of as 'global errors', which accumulate individual local errors over

time. The 10-day transport deviations are typically strongly affected by the individual atmospheric conditions, e. g., as particles enter chaotic regions and are dispersed by divergent flows.

The most important result of this analysis is that the transport deviations between ERA5 and ERA-Interim are substantially larger than the transport deviations due to diffusion and subgrid-scale wind fluctuations. After 1 day the transport deviations between ERA5 and ERA-Interim are up to an order of magnitude larger than the transport deviations due to diffusion and subgrid-scale wind fluctuations. After 10 days the differences are still larger by a factor of $2-3$. This indicates that there are considerable differences between Lagrangian transport simulations based on ERA5 and those based on ERA-Interim at all latitudes and in all height ranges considered here. Globally, the medians of the horizontal transport deviations at different height levels are in the range of $100-250\,\mathrm{km}$ (Fig. 8a) or $14-25\%$ (Fig. 8c) after 1 day and $1400-3500\,\mathrm{km}$ (Fig. 8b) or $16-35\%$ (Fig. 8d) after 10 days. The medians of the vertical transport deviations are in the range of $0.17-0.37\,\mathrm{km}$ (Fig. 8e) or $38-50\%$ (Fig. 8g) after 1 day and $0.5-1.4\,\mathrm{km}$ (Fig. 8f) or $14-19\%$ (Fig. 8h) after 10 days. The spatial differences between ERA5 and ERA-Interim trajectories are typically largest in the troposphere and in the middle to upper stratosphere, whereas ERA5 and ERA-Interim tend to agree best in the UT/LS region and the lower to middle stratosphere. A notable exception is the maximum in AVTD found in the UT/LS region in the tropics (Figs. 8e and 8f). In general, transport deviations in the middle and high latitudes of both hemispheres compare well to each other, but are distinctly different from those in the tropics. In particular, RHTDs in the tropics are larger than those in the extratropics (Figs. 8c and 8d). The largest peak-to-peak variations are mostly found in the middle and upper stratosphere (e. g., Figs. 8a and 8b), which indicates that annual cycles in the wind fields at these altitudes are represented differently in ERA5 and ERA-Interim.

One reason explaining the large differences between ERA5 and ERA-Interim in the troposphere and the tropical UT/LS region may be an improved representation of convective updrafts and other small-scale features due to better spatial resolution of the ERA5 data (cf. Fig. 2). To further assess the effect of convective updrafts and other types of vertical motion, we analyzed the total vertical displacements of particles seeded in the height range of $2-8\,\mathrm{km}$ along the 10-day trajectories. Figure 9 shows a 2-D histogram of the positive vertical displacements for June to August 2017 for the ERA5 trajectories, as well as the relative differences of this histogram with respect to ERA-Interim. Overall, the distribution of vertical displacements for the ERA5 trajectories looks realistic (Fig. 9a), as we would expect to find stronger updrafts associated with convection near the ITCZ and downdrafts or weaker updrafts in the subtropics due to the Hadley cells. A closer inspection of the relative differences (Fig. 9b) indicates that strong updrafts are found more frequently (up to 50%) in ERA5 compared with ERA-Interim in the extratropics. Stronger updrafts in ERA5 are associated with significantly larger vertical velocities (Fig. 9c). However, for the tropics the analysis shows that the number of strong updrafts is reduced (down to $-20\%$) in ERA5. This discrepancy may be due to the fact that the area in which strong tropical updrafts occur are more confined in ERA5 compared with ERA-Interim (compare Figs. 2c and 2d), such that fewer particles are affected by these updrafts. Convective properties are quite different in ERA5, which displays much more intermittency than ERA-Interim.

### 3.3 Tracer differences between ERA-Interim and ERA5 trajectories

In this section, we discuss the differences in meteorological variables and dynamical tracers sampled along the ERA5 and ERA-Interim trajectories. For temperature, we analyzed the mean absolute deviation (MAD). Specific humidity, potential temperature, and potential vorticity exhibit strong variations with height and are therefore compared using the mean relative deviation (MRD). The height ranges and latitude bands for the analysis are the same as before and the analysis covers the same global simulations for the year 2017. The results of the statistical analysis are presented in Fig. 10. Overall, this analysis confirms the key finding of Sect. 3.2 that there are substantial differences between Lagrangian transport simulations using ERA5 and those using ERA-Interim data. The deviations of the meteorological variables and dynamical tracers between ERA5 and ERA-Interim are significantly larger than those caused by parameterized diffusion and subgrid-scale wind fluctuations in all cases.

The medians of the global MADs of temperature are in the range of $0.7 - 3.0$ K after 1 day and $2 - 13$ K after 10 days (Figs. 10a and 10b), with smallest values found in the lower and middle stratosphere and the largest values found in the troposphere. Temperature MADs in the extratropics are quite similar to global values. In contrast, temperature MADs in the tropics are largest in the UT/LS region, which correlates with particularly large AVTDs in this region (see Figs. 8e and 8f). For specific humidity we found median global MRDs of 29% in the troposphere, 26% in the UT/LS region, and $\leq 4\%$ in the stratosphere after 1 day (Fig. 10c). After 10 days, the MRDs increase to 85% in the troposphere and 45% in the UT/LS region, but still remain below 5% in the stratosphere (Fig. 10d). The large differences between the ERA5 and ERA-Interim specific humidities in the troposphere and UT/LS region are associated with large variability of specific humidity itself in these regions. The stratosphere is very dry and exhibits much lower variations in specific humidity compared with the troposphere. However, the small stratospheric differences reported here are significant in comparison to those arising from diffusion and subgrid-scale wind fluctuations (see also Fig. 5d). As for temperature, the largest relative differences between ERA5 and ERA-Interim specific humidity are found in the troposphere in the extratropics and in the UT/LS region in the tropics, and can be traced back to the respective AVTDs.

Turning to the dynamical tracers, global median MRDs of potential temperature are in the range of $0.4 - 1.6\%$ after 1 day and $1.4 - 5.2\%$ after 10 days (Figs. 10e and 10f). MRDs of potential temperature mostly increase with height, in particular in the stratosphere. This is partially related to the exponential increase of potential temperature with height, which is not entirely suppressed by analyzing relative rather than absolute deviations. For the second dynamical tracer, potential vorticity, we found much larger deviations between ERA5 and ERA-Interim (Figs. 10g and 10h). Global median MRDs in potential vorticity after 1 day are about 50% in the troposphere and UT/LS region and around $16 - 24\%$ in the stratosphere. MRDs in all four altitude ranges further increase to $20 - 80\%$ after 10 days. The largest MRDs are found in the tropics, which might be due to the fact that values of potential vorticity in this region are small when compared with those in the extratropics. Overall, the rather large deviations of potential vorticity between ERA5 and ERA-Interim were surprising. Additional tests showed that these differences are comparable when we use the CLaMS model instead of the MPTRAC model for this analysis, and that they are much larger than differences between the two models (see Sect. 3.6). A possible reason for the large relative deviations is

that ERA5 exhibits more fine structure in the potential vorticity fields than ERA-Interim, because of its better resolution (cf. Figs. 2e and 2f). Differences in vertical dispersion may also play a role, given the relatively large vertical gradient of potential vorticity around the tropopause.

Next to MADs and MRDs, which measure variability between the ERA5 and ERA-Interim tracer data along the trajectories, we also analyzed for biases, which measure the systematic differences between the means of the distributions. The results of this statistical analysis are presented in Fig. 11. Overall, the biases are notably smaller than the MADs or MRDs, typically by a factor of 2 or more. However, in nearly all cases the biases are larger than the systematic differences introduced by parameterized diffusion and subgrid-scale wind fluctuations. Global temperature biases of ERA5 minus ERA-Interim are in the range of $-0.2$ to $1.3$ K, with the largest positive biases being found in the mid to upper stratosphere after 1 day (Fig. 11a) and in the troposphere after 10 days (Fig. 11b). This bias along the trajectories is partly due to direct biases between ERA5 and ERA-Interim temperature data (see Figs. 3d and 4d in Sect. 2.1.2). Global relative biases of specific humidity remain in the range of $-18$ to $6\%$ after 10 days (Fig. 11d). Significantly smaller specific humidities of ERA5 compared to ERA-Interim in the UT/LS region already after 1 day seem noteworthy (Fig. 11c), as they can be attributed to direct biases between the data sets in this region (see Figs. 3e and 4e in Sect. 2.1.2). Being correlated with temperature biases, global relative biases of potential temperature remain in the range of $-0.4\%$ in the troposphere to $0.9\%$ in the mid to upper stratosphere after 10 days (Fig. 11f). Global relative biases of potential vorticity are in the range of $-4$ to $8\%$ after 10 days (Fig. 11h). A systematic, yet unexplained difference in potential vorticity between the Southern Hemisphere and Northern Hemisphere extratropics became evident in the troposphere and UT/LS region already after 1 day (Fig. 11g).

## 3.4 Tracer conservation along ERA5 and ERA-Interim trajectories

Direct validation of trajectory calculations can be performed by means of comparison to balloon observations (e. g., Knudsen and Carver, 1994; Baumann and Stohl, 1997; Hertzog et al., 2004; Riddle et al., 2006; Friedrich et al., 2017; Hoffmann et al., 2017). However, this type of validation is limited by the sparse spatial and temporal coverage of the balloon data. In this study, we followed the approach of Stohl and Seibert (1998) by conducting a systematic global assessment of our trajectory calculations with respect to the conservation of dynamical tracers along trajectories, including specific humidity, potential temperature, and potential vorticity. We performed this analysis for both ERA5 and ERA-Interim to assess whether tracer conservation has improved in ERA5. The results are summarized in Fig. 12.

Conservation of specific humidity applies unless the parcel is affected by condensation, evaporation, chemical reactions, or mixing (Gray et al., 1994; Salathé Jr and Hartmann, 1997; Röckmann et al., 2004; Galewsky et al., 2005). In the free troposphere, specific humidity can be considered to be a dynamical tracer on short timescales, such as a few hours to a day. In the stratosphere, even longer timescales apply. In our simulations, we found global RTCEs of specific humidity of about $30\%$ in the troposphere and $20\%$ in the UT/LS region after 1 day (Fig. 12a). These results compare well to those reported by Stohl and Seibert (1998), who found a specific humidity RTCE of about $35\%$ after 24 h for 3-dimensional tropospheric trajectories calculated using ECMWF meteorological data. Stratospheric values of the RTCE are very low ($\leq 2\%$), because of better conservation and the weak spatiotemporal variability of specific humidity itself in this region. RTCEs of specific humidity

exhibit some variations with latitude, in particular in the troposphere and in the UT/LS region. The largest conservation errors are in the troposphere in the extratropics whereas they maximize in the UT/LS in the tropics. RTCEs in tropospheric specific humidity are quite similar between ERA5 and ERA-Interim. After 10 days RTCEs in the troposphere exceed 100% (Fig. 12b), at which point we may confidently say that conservation of specific humidity no longer applies. Tracer conservation errors in

the UT/LS region rise to 30% in the extratropics and 100% in the tropics after 10 days, although stratospheric RTCE values remain well below 5%.

Potential temperature and potential vorticity are conserved in reversible adiabatic processes and will not change in the absence of heating, cooling, evaporation, condensation, or mixing (e. g., Curry, 2015; McIntyre, 2015). Our analysis of tracer conservation for potential temperature revealed major improvements when the new ERA5 products are used in place of ERA-

10 Interim throughout the stratosphere and UT/LS. Global median RTCEs of potential temperature after 1 day are in the range of $0.4 - 1.6\%$ for ERA-Interim, but as low as $0.2 - 0.6\%$ for ERA5 (Fig. 12c). After 10 days, RTCE values increase to $1.9 - 6.2\%$ for ERA-Interim and $1.8 - 4.5\%$ for ERA5 (Fig. 12d). RTCEs for potential temperature are quite similar among the different latitude bands. Following Schoeberl (2004), Fig. 13 further illustrates the improvements in consistency and tracer conservation of potential temperatures for ERA5. The figure shows the dispersion of 10-day trajectories from seeds at potential temperature

levels ranging from 400 to 1200 K for simulations initialized on 1 July 2017. The results for both data sets reveal downwelling of air in the Southern Hemisphere polar vortex and upwelling over the ITCZ. However, much larger dispersion or 'scattering' of the final positions of the trajectories is found in the simulations based on ERA-Interim relative to those based on ERA5, especially above the 800 K isentropic surface. Possible reasons for improved conservation of potential temperatures in simulations based on ERA5 compared to those based on ERA-Interim may be improved internal consistency of the ECMWF forecast

model or between the model and observations as well as shorter analysis intervals, leading in turn to smaller assimilation increments in temperature.

We found much larger tracer conservation errors for potential vorticity than for potential temperature. Global median RTCEs are in the range of $48 - 54\%$ in the troposphere, $44 - 48\%$ in the UT/LS region, and $8 - 18\%$ in the stratosphere after 1 day (Fig. 12e). The stratospheric values compare well to estimates of relative potential vorticity changes calculated for balloon trajec-

25 tories by Knudsen and Carver (1994), whereas the tropospheric values are about $10 - 20$ percentage points larger than those reported by Stohl and Seibert (1998). After 10 days the RTCEs increased to $90 - 100\%$, $60 - 70\%$, and $20 - 50\%$, respectively, in the same three height ranges (Fig. 12f). We found that tracer conservation is similar or slightly improved when using ERA5 data in the stratosphere, but it is weaker in the troposphere and UT/LS region. Following Stohl and Seibert (1998), we conducted several tests to check whether RTCEs can be improved by excluding trajectories for which potential vorticity conservation is

30 not likely to be applicable. We excluded trajectories entering levels below 1 km altitude above the surface, to avoid turbulent and unstable conditions in the planetary boundary layer. We also excluded trajectories with relative humidities larger than 90%, as condensation or evaporation may cause diabatic temperature changes in such cases. However, these tests did not yield any substantial changes in our RTCE results. The increase in tropospheric RTCEs of potential vorticity between ERA-Interim and ERA5 might be due to higher spatiotemporal resolution in ERA5, which allows for finer structures in the potential vorticity

fields relative to ERA-Interim (see Sect. 3.3). The small improvements in stratospheric RTCEs are likely related to improved conservation of potential temperature along trajectories.

## 3.5 Downsampling experiments with ERA5

As spatial and temporal resolution is a key factor in the trade-off between accuracy and computational time of Lagrangian transport simulations (Stohl et al., 1995; Stohl and Seibert, 1998; Pisso et al., 2010; Bowman et al., 2013), our study covers a number of downsampling experiments using ERA5 data. The process of downsampling or decimation to reduce the sampling rate of a signal typically consists of two steps (e. g., Lyons, 2010). The first step is to apply a low pass filter to the original data to avoid aliasing of high-frequency features. Here, we applied smoothing with triangular weights in space and time to achieve this effect. The second step is to subsample the smoothed data on the reduced grid. For example, to downsample ERA5 data from hourly to 2-hourly time intervals, we averaged data of $\{t - 1\,\mathrm{h}, t, t + 1\,\mathrm{h}\}$ for a given time $t$ with weighting factors of $\{0.25, 0.5, 0.25\}$ and kept the smoothed data only at a 2-hourly interval. Sensitivity tests showed that this approach including low-pass filtering may significantly reduce aliasing errors and improve simulation results.

We conducted four downsampling experiments with the ERA5 data, in which we reduced (I) the number of synoptic time steps $n_t$ by a factor of 2, (II) the number of vertical levels $n_{lev}$ by a factor of 2, (III) the numbers of longitudes $n_{lon}$ and latitudes $n_{lat}$ by a factor of 2, and (IV) $n_t$ by a factor of 6, $n_{lev}$ by a factor of 2, and $n_{lon}$ and $n_{lat}$ by a factor of 3. Experiment IV was set up to achieve a spatiotemporal sampling similar to ERA-Interim. In order to enable a fair comparison, in Experiment IV the low-pass filtering in the temporal domain was switched off and only subsampling was applied, as both ERA5 and ERA-Interim winds are instantaneous values rather than time-integrated quantities. We quantified the differences of the Lagrangian transport simulations using the downsampled and the full resolution ERA5 data by calculating transport deviations after 1 day, as these are most sensitive to the specific uncertainties and less dependent on the individual meteorological conditions and flow conditions (Rößler et al., 2018). Figures 14 and 15 show the results of these four experiments.

Considering the downscaling experiments I – III (Fig. 14), it was found that the impacts of downsampling of the ERA5 data are comparable to the impacts of parameterized diffusion and subgrid-scale wind fluctuations in most cases. The impacts of downsampling generally tend to be strongest in the troposphere, where transport deviations due to downsampling exceed those by diffusion and subgrid-scale wind fluctuations by up to a factor of 3. In the UT/LS region the horizontal transport deviations exceed those by diffusion and subgrid-scale wind fluctuations by up to a factor of 2 (Figs. 14a and 14b), whereas the vertical transport deviations are smaller by up to a factor of 2 (Figs. 14c and 14d). For the stratosphere the experiments suggest that we can downsample from hourly to 2-hourly data or that we can reduce the horizontal sampling by a factor of $2 \times 2$ without any significant impact compared to diffusion and subgrid-scale wind fluctuations. This may reflect the reduced sensitivity of the stratosphere to downsampling in the horizontal direction and in time, as the stratosphere is dynamically more stable and has a redder spectrum of motion than the troposphere. The number of vertical levels $n_{lev}$ should not be reduced in the stratosphere, because the vertical sampling even of the high-resolution ERA5 data is relatively coarse at stratospheric levels (see Fig. 1).

Downsampling experiment IV (Fig. 15) is intended to separate the impact of improved spatiotemporal resolution from the impacts of other improvements from ERA-Interim to ERA5, such as modified physical parameterizations in the forecast model

or improved data assimilation procedures and observations. For this reason, transport deviations between the downsampled and full-resolution ERA5 data are compared to transport deviations between ERA5 and ERA-Interim and not with diffusion and subgrid-scale wind fluctuations in Fig. 15. In this experiment we found that transport deviations between simulations based on downsampled and full-resolution ERA5 data are mostly smaller than the deviations between ERA-Interim and ERA5. This indicates that the transport deviations between ERA-Interim and ERA5 as discussed in Sect. 3.2 are due to both, improved resolution in ERA5 as well as other improvements in the forecast model and data assimilation scheme, and cannot be attributed to a single cause. Vertical transport deviations in the stratosphere are an exception, as the deviations due to downsampling became larger than the deviations between ERA-Interim and ERA5. Aliasing effects play a strong role in this case, as the vertical transport deviations in the stratosphere are reduced by a factor of $3-4$ if low-pass filtering is taken into account. Other transport deviations are less affected by temporal low-pass filtering. In summary, using downsampled ERA5 data should generally not be considered to be equivalent to using ERA-Interim data for Lagrangian transport simulations.

### 3.6 Comparison of the CLaMS and MPTRAC models

Finally, we conducted a comparison of Lagrangian transport simulations using two different models, CLaMS and MPTRAC. This allows us (i) to check the consistency of the model results and (ii) to assess the readiness of both models for operating with the comprehensive ERA5 data set. The necessary adjustments in the codes and workflows for both models to make use of ERA5 data have been described in Appendix A. In this comparison, we focus on global transport deviations as well as differences in meteorological variables and dynamical tracers between CLaMS and MPTRAC after 1 day of integration at different height ranges. All simulations for the year 2017 are included. The results are shown in Fig. 16.

Overall, the model comparison revealed excellent agreement between CLaMS and MPTRAC kinematic trajectory calculations using ERA5 data. Transport deviations between the models are significantly smaller than those due to parameterized diffusion and subgrid-scale wind fluctuations in the MPTRAC model in most cases (Figs. 16a to 16d). The only notable exception is horizontal transport deviations in the middle and upper stratosphere (Fig. 16a), which are similar to or slightly exceed the deviations due to diffusion and subgrid-scale wind fluctuations. We have tested whether these differences are due to the different vertical interpolation schemes applied in the models, with CLaMS using logarithmic interpolation and MPTRAC using linear interpolation with respect to pressure, but found that this has only marginal impact. Furthermore, the results are robust against changes in the time step applied in the MPTRAC model. Nevertheless, the global AHTDs (RHTDs) between CLaMS and MPTRAC are less than 9 km (1.5%) from the troposphere to the middle stratosphere and less than 30 km (2.3%) in the middle and upper stratosphere at all latitudes. The global AVTDs (RVTDs) are less than 40 m (6%) at all heights.

In most cases, transport deviations between CLaMS and MPTRAC do not lead to large deviations in meteorological variables or dynamical tracers sampled along the trajectories (Figs. 16e to 16h). Temperature MADs are less than 0.25 K, specific humidity MRDs below 2.2%, and potential temperature MRDs are less than 2.0%. Larger differences (up to $12-13\%$) were found for potential vorticity in the troposphere and UT/LS region. This may reflect the fact that numerical calculations of potential vorticity are particularly sensitive to fine-scale structure and variability in the horizontal wind field in this part of the

atmosphere (see Sect. 3.3). In the stratosphere, differences in potential vorticity between CLaMS and MPTRAC simulations are comparable to or smaller than transport deviations due to diffusion and subgrid-scale wind fluctuations.

## 4 Discussion and conclusions

In this study, we have assessed the impact of ECMWF's next-generation ERA5 reanalysis on Lagrangian transport simulations
and quantified some of the differences with respect to the well-established and widely used ERA-Interim reanalysis. To quantify the impact of the new ERA5 data, we conducted global simulations for the free troposphere and stratosphere for the year 2017, each covering 24 sets of 10-day forward trajectories. Based on a comprehensive statistical analysis of transport deviations, we concluded that the new ERA5 data have considerable impact on Lagrangian transport simulations. Transport deviations (AHTDs and AVTDs) indicating differences between ERA5 and ERA-Interim are up to an order of magnitude larger than
those caused by parameterized diffusion and subgrid-scale wind fluctuations after 1 day and still up to a factor of $2-3$ larger after 10 days. Depending on the height range, spatial differences between trajectories using ERA5 and those using ERA-Interim map into global differences of up to 3 K in temperature, 30% in specific humidity, 1.8% in potential temperature, and 50% in potential vorticity after only 1 day of integration. These differences are much larger than those due to numerical errors in the trajectory calculations (e. g., Rößler et al., 2018) and those between the different Lagrangian models CLaMS and MPTRAC.
Monthly mean zonal mean temperatures and zonal winds were found to be in good agreement between ERA5 and ERA-Interim, except for some differences in the upper stratosphere, where ERA5 has substantially finer vertical resolution than ERA-Interim. However, direct comparison of horizontal wind, vertical velocity, and potential vorticity maps for the troposphere and an example of trajectory calculations for the stratosphere revealed more detailed fine structures in ERA5 in comparison to ERA-Interim. These fine structures are associated with the better spatial and temporal resolution of ERA5 data. In the
troposphere, we found stronger updrafts in the extratropics and a more realistic representation of tropical cyclones in ERA5 relative to ERA-Interim, which are partly related to the improved spatiotemporal resolution offered by ERA5. However, fewer strong updrafts are found in the tropics in ERA5, which may have important implications for the distribution of water vapor in the UT/LS region and the lower stratosphere. For the stratosphere, we found that the conservation of potential temperature along the trajectories is significantly improved when the new ERA5 data are used in place of ERA-Interim products. This may
be due to better consistency between ECMWF's forecast model and observations and shorter analysis cycles yielding smaller data assimilation increments.

Compared with ERA-Interim, the new ERA5 reanalysis incorporates a decade of research on forecast modeling, observational systems, and data assimilation. Although there are many changes and improvements from ERA-Interim to ERA5, the impact of the new reanalysis on Lagrangian transport simulations and other applications still needs to be further assessed.
In this study, we have focused on quantifying the differences between the trajectories based on ERA5 and those based on ERA-Interim in terms of dynamical tracer conservation. Future work may focus on direct validation of the new ERA5 products via comparison with independent observations. Another interesting aspect is that ERA5 provides information on uncertainty through a 10-member ensemble of data assimilations, which could be taken into account in future studies (e. g., by means of

ensemble trajectory simulations). The total amount of data associated with the ECMWF reanalyses has increased by a factor of ∼80 from ERA-Interim in 2006 to ERA5 in 2016, whereas the capacity of hard disks, measured in terms of areal density, grew only by a factor of ∼10 per decade during that time (Freitas et al., 2011). Downsampling to reduce the amount of data can be an option for applications that require only coarser resolution. However, many Lagrangian transport models and chemistry-transport models will need careful code optimization and tuning to cope with the 'big data' challenge presented by ERA5, and to fully realize the benefits of ERA5 data at its full resolution.

*Code and data availability.* We retrieved ERA5 and ERA-Interim reanalysis data (Dee et al., 2011; Hersbach and Dee, 2016) from the European Centre for Medium-Range Weather Forecasts (ECMWF) Meteorological Archival and Retrieval System (MARS). ECMWF data have been processed for usage with MPTRAC by means of the Climate Data Operators (Schulzweida, 2014). The MPTRAC model (Hoffmann et al., 2016) is freely available under the terms and conditions of the GNU General Public License, version 3, from the repository at https://github.com/slcs-jsc/mptrac (last access: 14 November 2018). The box model version (trajectory module including chemistry) of CLaMS (McKenna et al., 2002a, b) is also available and can be obtained by contacting Rolf Müller, Jülich.

## Appendix A: Simulation workflows

We had to change the typical workflows for the Lagrangian transport simulations in this study, mainly because of the large volume of the ERA5 data and the computational resources required to handle it. Primarily, the ERA5 and ERA-Interim data are stored in ECMWF's main repository of meteorological data, the Meteorological Archival and Retrieval System (MARS), which is accessible by means of a web interface and more recently, via the Copernicus Climate Change Service (C3S) Climate Data Store (CDS). The C3S CDS is the favored pathway for the distribution of ERA5 data and expected to become the only source of ERA5 data in the future. However, the retrieval of ECMWF data on both pathways, C3S CDS and MARS, is not designed to be instant. Requests for a large amount of data can take days to weeks to complete. For Lagrangian transport simulations and various other applications, the data must be transferred and archived locally at a computing site, before they can be used effectively.

At the Jülich Supercomputing Centre different user groups traditionally maintained their own archives of meteorological data. However, considering the volume of the ERA5 data, the approach of having multiple copies of the same data is no longer considered justifiable. Therefore, a joint meteorological data archive was established, referred to as the 'meteocloud', to store large reanalysis and satellite data sets. The meteocloud archive is made accessible to local users of the facility for scientific collaboration. A survey was conducted to identify the specific variables of the ERA5 data needed by different user groups for their research applications. Data for those variables are retrieved from the ECMWF main repository in gridded binary (grib) format and stored on a dedicated shared disk space with fast access. At present, the meteocloud archive has a capacity of nearly 600 TByte of disk space, which will be sufficient to store more than two decades of ERA5 data.

The implementation of the meteocloud archive required changes in the workflows for the Lagrangian transport model simulations. For example, the preprocessing of meteorological input data for use with the MPTRAC model was integrated directly

into the workflow. We implemented a simple mechanism that can be used for 'staging' of meteorological input data during the course of a simulation. While the model is running, the staging mechanism steadily checks, whether the required meteorological input files for MPTRAC are available for the given time step. In case of missing input data, it triggers an external script to convert the ERA5 grib files retrieved from ECMWF to the specific binary format needed by MPTRAC. The MPTRAC input files are saved on a scratch storage volume, where they remain as long as free disk space is available. Running multiple simulations with the same input data may thus benefit from a caching effect. The implementation of this staging mechanism was rather simple, because we had to apply only minimal changes to the file input routines of the MPTRAC model. For the CLaMS model another optimization of the file input routines was implemented, so that only spatial subsets of the full global meteorological data fields were read in as needed. We found both methods to be effective adaptations of the codes and workflow that enable CLaMS and MPTRAC models to cope with the large amount of ERA5 data.

*Author contributions.* LH developed the concept for this study and conducted the formal analysis of the results. LH and DL carried out the MPTRAC and CLaMS simulations. GG and OS were responsible for curation of the ECMWF reanalyses data. XW compiled the overview of the meteorological conditions during the year 2017. LH prepared the manuscript with contributions from all co-authors.

*Competing interests.* The authors declare that they have no conflict of interest.

*Acknowledgements.* ERA5 data were generated using Copernicus Climate Change Service Information. Neither the European Commission nor ECMWF are responsible for any use that may be made of the Copernicus information or data in this publication. We acknowledge the Jülich Supercomputing Centre for providing computing time on the supercomputer JURECA and for storage resources for the meteocloud data archive. Yi Heng acknowledges support provided by the Thousand Talents Program for Young Scholars of China and was supported by the Natural Science Foundation of Guangdong (China) under grant no. 2018A030313288. Dan Li was supported by the International Postdoctoral Exchange Fellowship Program 2017 under grant no. 20171015. Xue Wu was supported by the National Natural Science Foundation of China under grant no. 41605023 and the China Postdoctoral Science Foundation under grant no. 2018T110131.

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

**Table 1.** Characteristics of the ERA5 and ERA-Interim reanalyses as well as resource requirements to calculate 10-day forward trajectories for $10^6$ particles with the MPTRAC model on a single compute node (including 24 cores) of the supercomputer JURECA in Jülich.

|  | ERA5 | ERA-Interim |
| --- | --- | --- |
| Characteristics |  |  |
| Implementation date | 8 March 2016 | 12 December 2006 |
| Horizontal resolution | $T_L 636$ ($\sim$31 km) | $T_L 255$ ($\sim$79 km) |
| Horizontal transform grid[a] | $0.3° \times 0.3°$ | $0.75° \times 0.75°$ |
| Vertical resolution | 137 levels up to 0.01 hPa | 60 levels up to 0.1 hPa |
| Temporal resolution | hourly | 6-hourly |
| IFS Cycle[b] | 41r2 | 31r2 |
| Period covered | 1950 – now | 1979 – now |
| Reference | (Hersbach and Dee, 2016) | (Dee et al., 2011) |
| Resource requirements |  |  |
| CPU-time [s] | 3130 | 350 |
| Main memory [MB] | 5800 | 530 |
| Disk storage [GB] | 450 | 5.8 |

[a]) These entries refer to the longitude $\times$ latitude grids at which we retrieved the data from ECMWF.

[b]) For a detailed description of ECMWF's Integrated Forecast System (IFS) cycle characteristics see

https://www.ecmwf.int/en/forecasts/documentation-and-support/changes-ecmwf-model (last access: 14 November 2018).

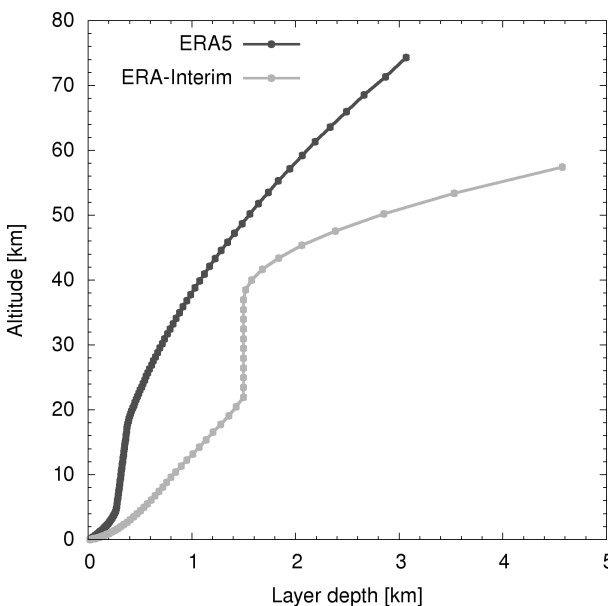

**Figure 1.** Vertical coverage and sampling of the ERA-Interim (light gray) and ERA5 (dark gray) reanalyses. Shown are layer depths and mid-layer altitudes calculated by means of the barometric formula using a constant scale height of 7 km and a surface pressure of 1013.25 hPa.

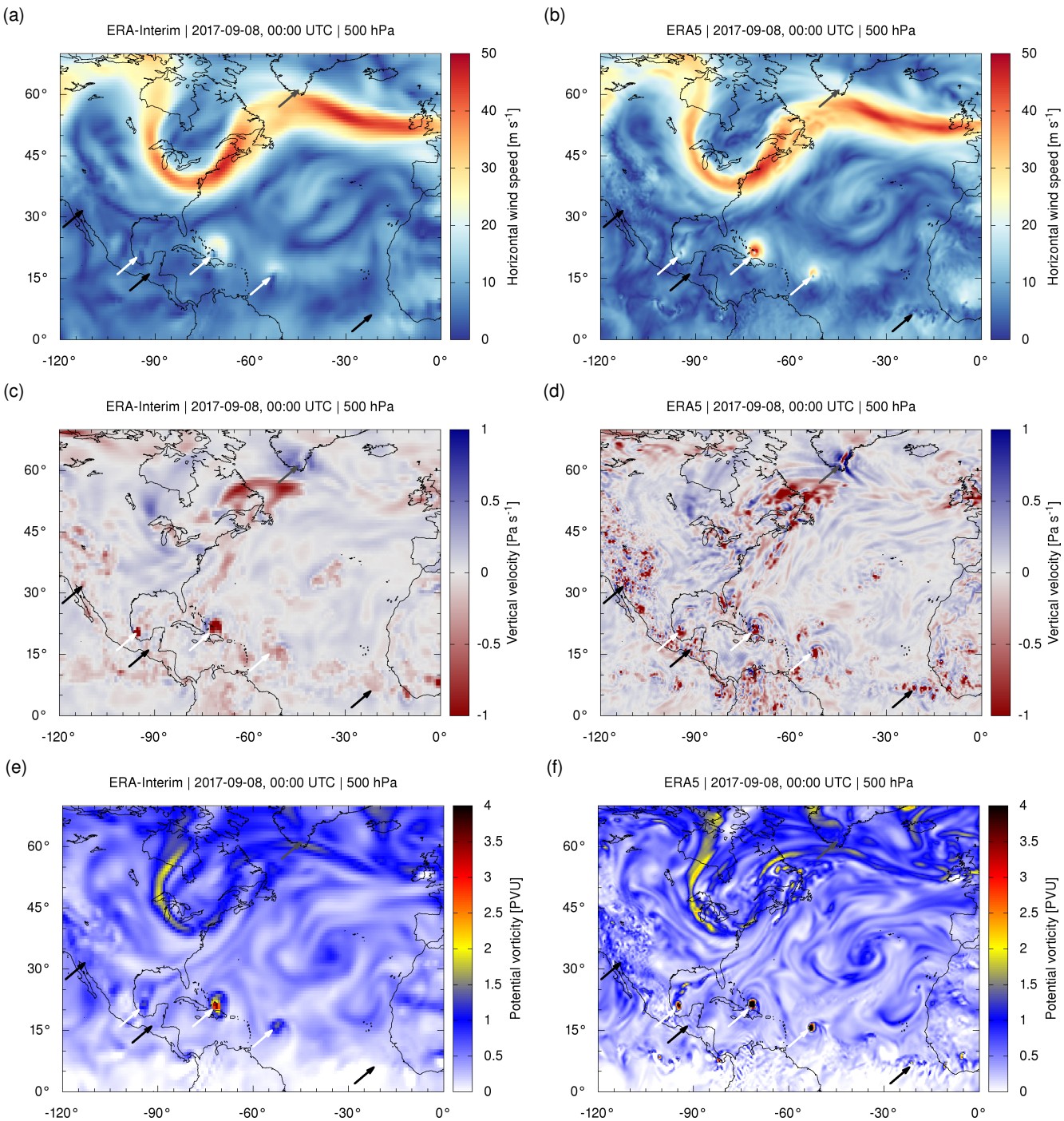

**Figure 2.** Comparison of ERA-Interim (left) and ERA5 (right) horizontal wind speeds (top), vertical velocities (middle), and potential vorticities (bottom) on 8 September 2017, 00:00 UTC over North America and the North Atlantic. Maps refer to the 500 hPa level (about 5 km of altitude). Arrows are used to point out the hurricanes Katia, Irma, and Jose (white, from west to east) as well as examples of gravity waves (gray) and explicitly resolved convective updrafts (black).

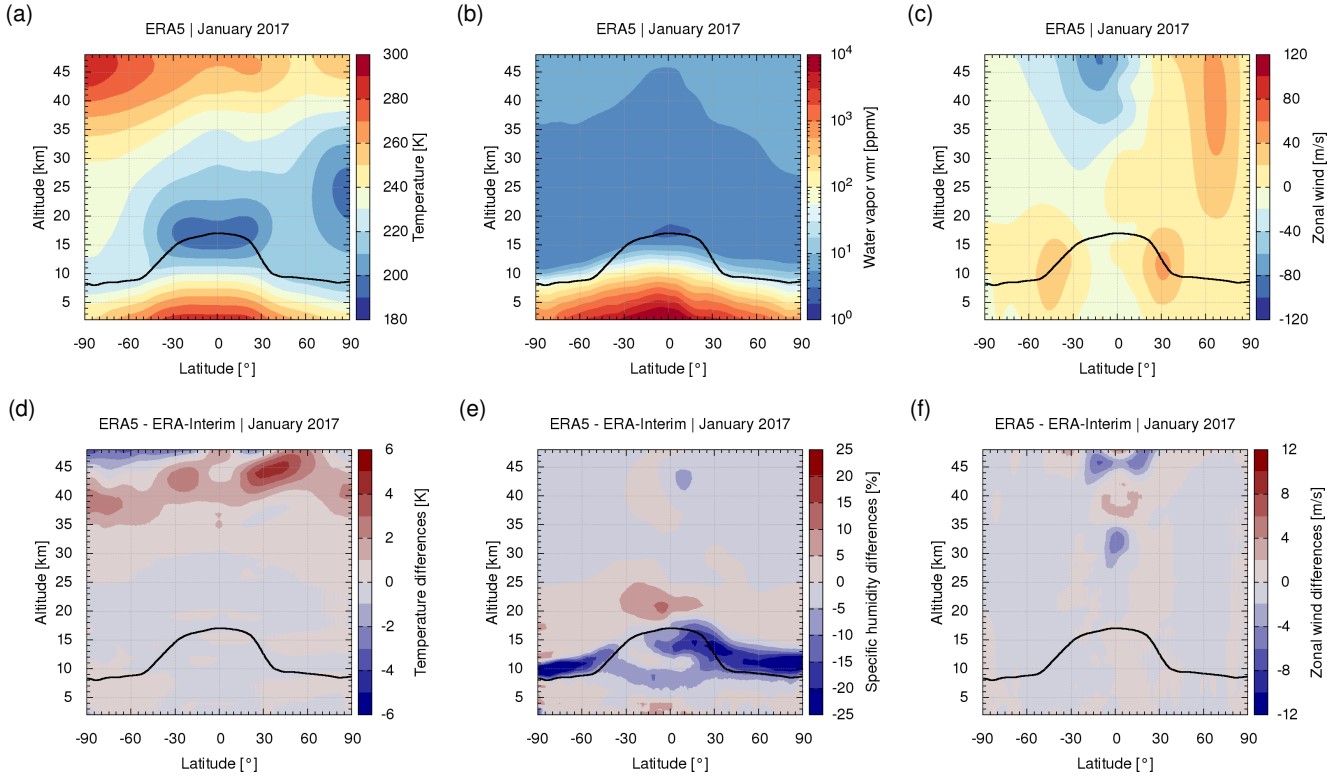

**Figure 3.** Zonal mean temperatures, water vapor volume mixing ratios, and zonal winds based on ERA5 (top) as well as corresponding differences between ERA5 and ERA-Interim (bottom) in January 2017. The black curve shows the zonal mean log-pressure height of the dynamical tropopause (based on thresholds of 3.5 PVU at mid and high latitudes and 380 K in the tropics).

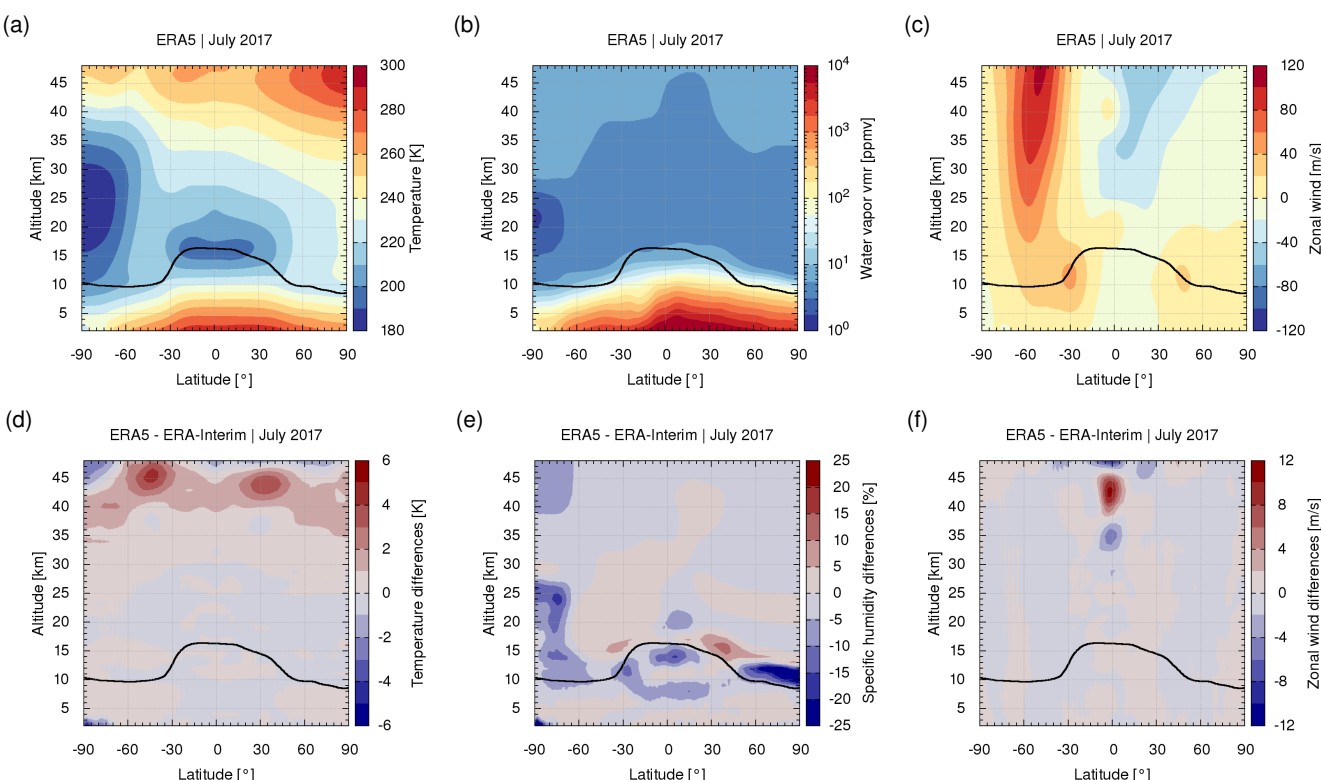

**Figure 4.** Same as Fig. 3, but for July 2017.

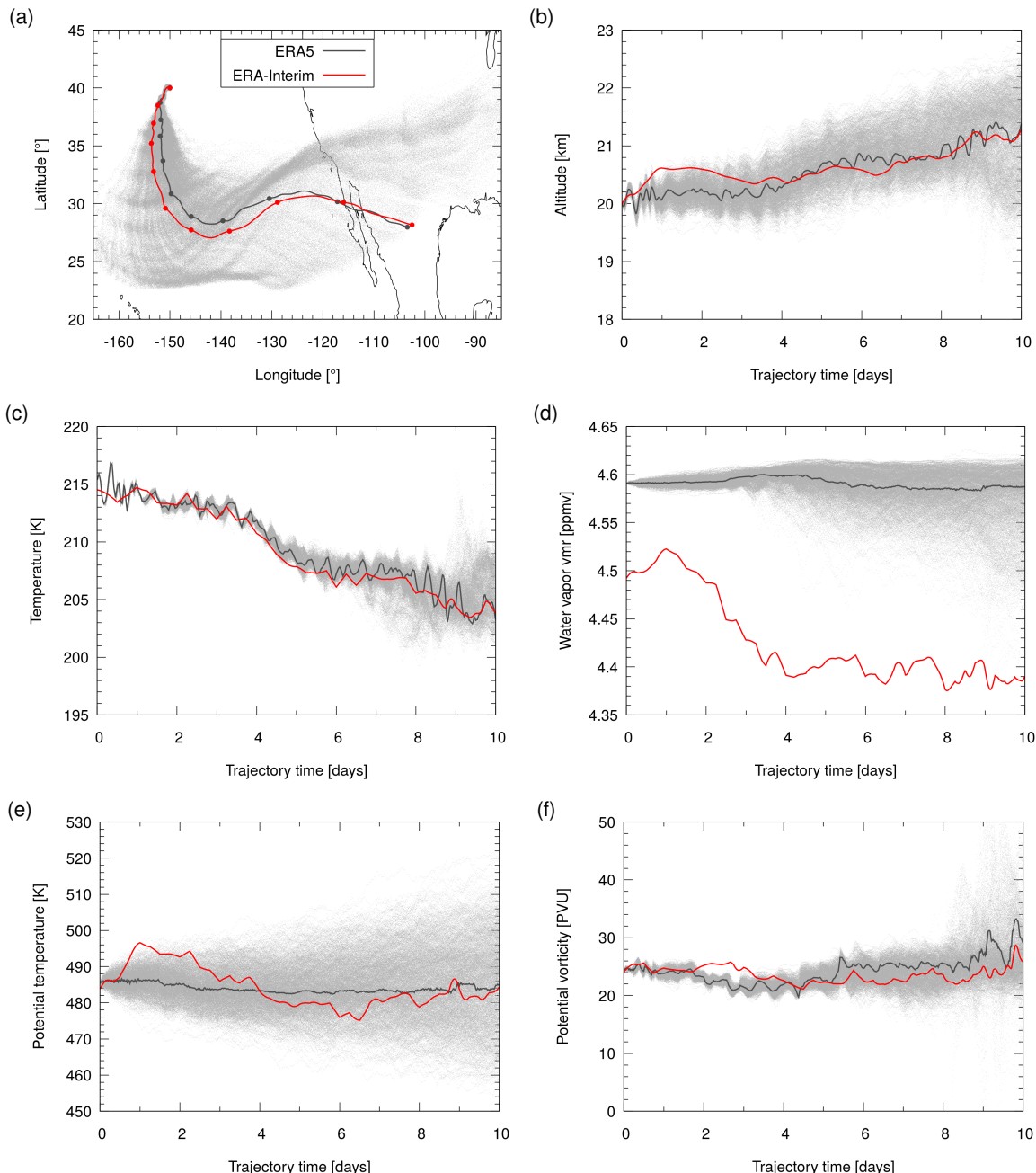

**Figure 5.** Particle positions (top), meteorological variables (middle), and dynamical tracers (bottom) sampled along a 10-day forward trajectory calculated with either ERA-Interim (red) or ERA5 (dark gray). Also shown is a 1000-member set of ERA5 trajectories with additional modeling of diffusion and subgrid-scale wind fluctuations (light gray). All trajectories have been launched on 1 January 2017, 00:00 UTC at (40°N, 150°W) and 58.2 hPa (about 20 km of altitude). The model output was saved every 20 min. Bullet points in (a) indicate 24 h intervals.

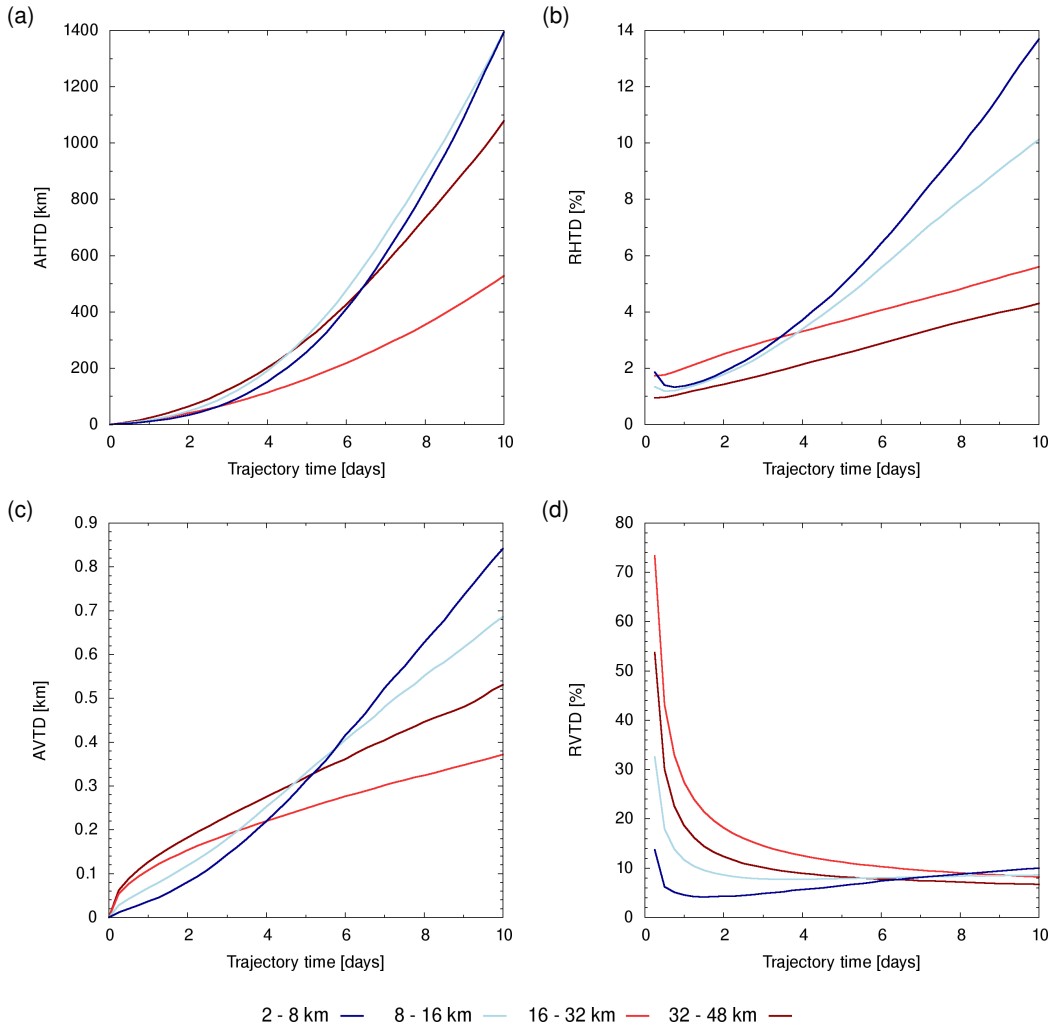

**Figure 6.** Global horizontal (top) and vertical (bottom) transport deviations of 10-day forward trajectories due to parameterized diffusion and subgrid-scale wind fluctuations. All trajectories were launched on 1 July 2017, 00:00 UTC and calculated with the MPTRAC model driven by ERA5 data. The color coding refers to different altitude ranges.

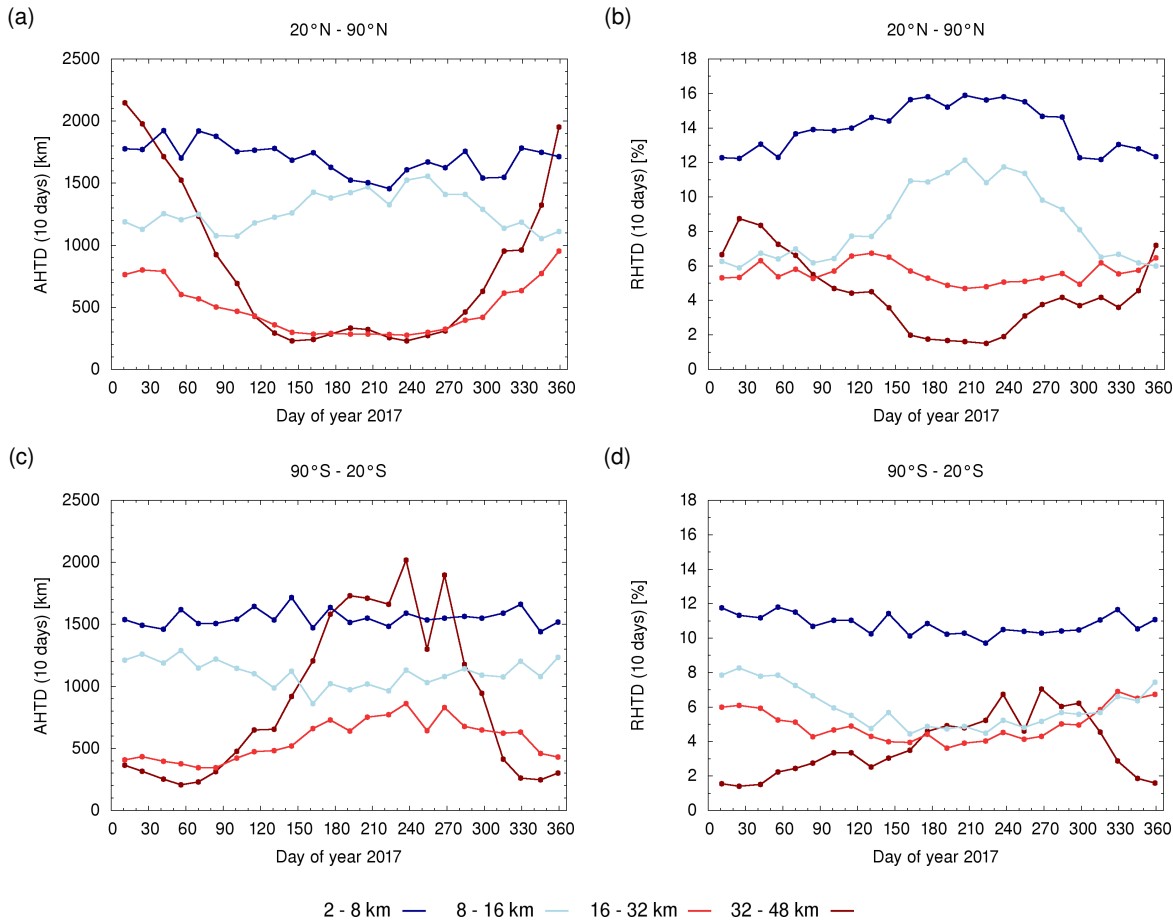

**Figure 7.** Seasonal variations of absolute (left) and relative (right) horizontal transport deviations due to parameterized diffusion and subgrid-scale wind fluctuations after 10 days of simulation time for the Northern Hemisphere (top) and Southern Hemisphere (bottom) extratropics. Trajectories were calculated with ERA5 data and launched at 00:00 UTC on the 1st and 15th of each month in 2017. The color coding refers to different altitude ranges.

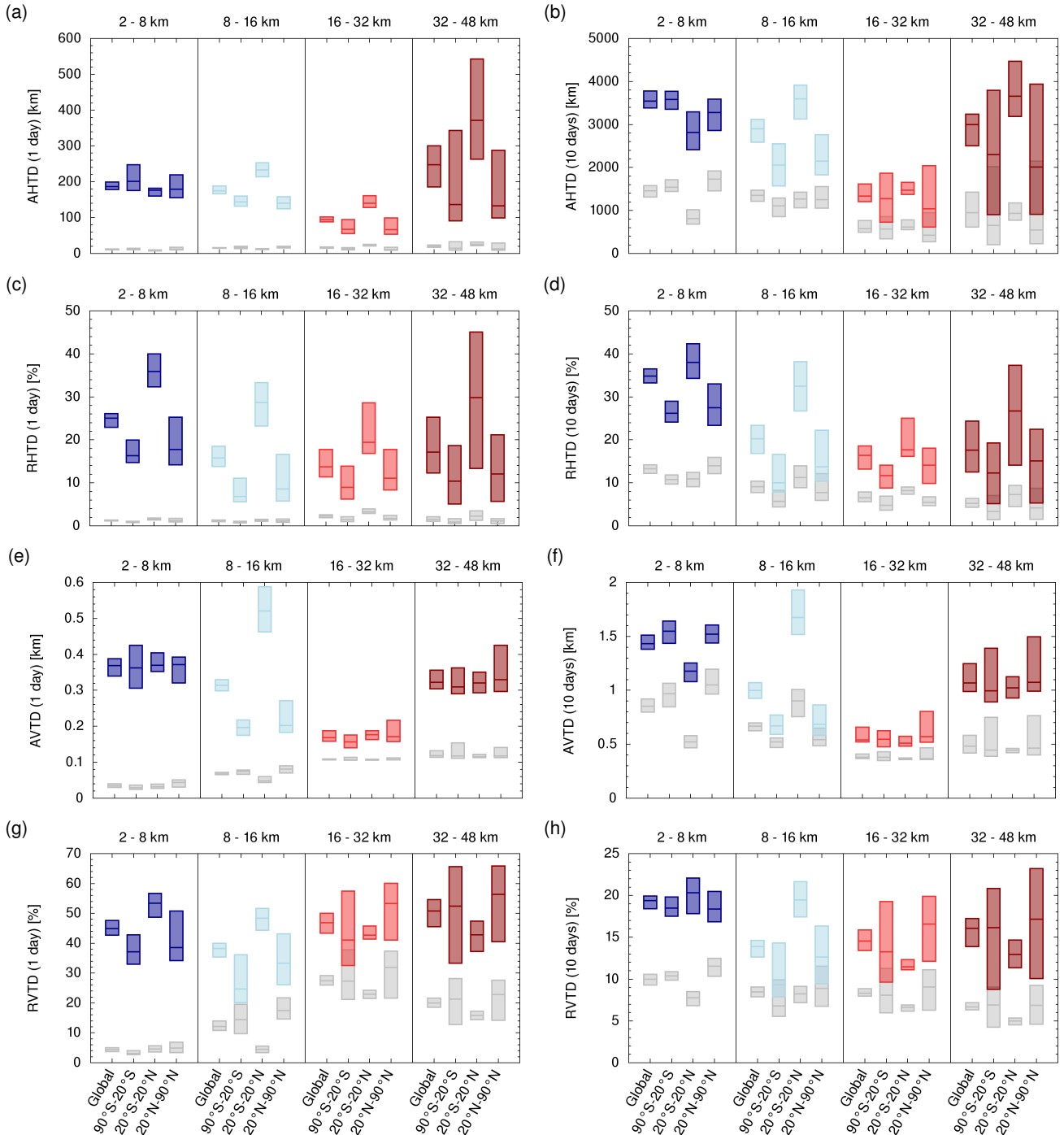

**Figure 8.** Transport deviations between ERA-Interim and ERA5 forward trajectories (blue and red bars for different height ranges) and transport deviations due to parameterized diffusion and subgrid-scale wind fluctuations (corresponding light gray bars) after 1 day (left) and 10 days (right) of time. The bars indicate the peak-to-peak range and the median of 24 trajectory simulations covering the year 2017.

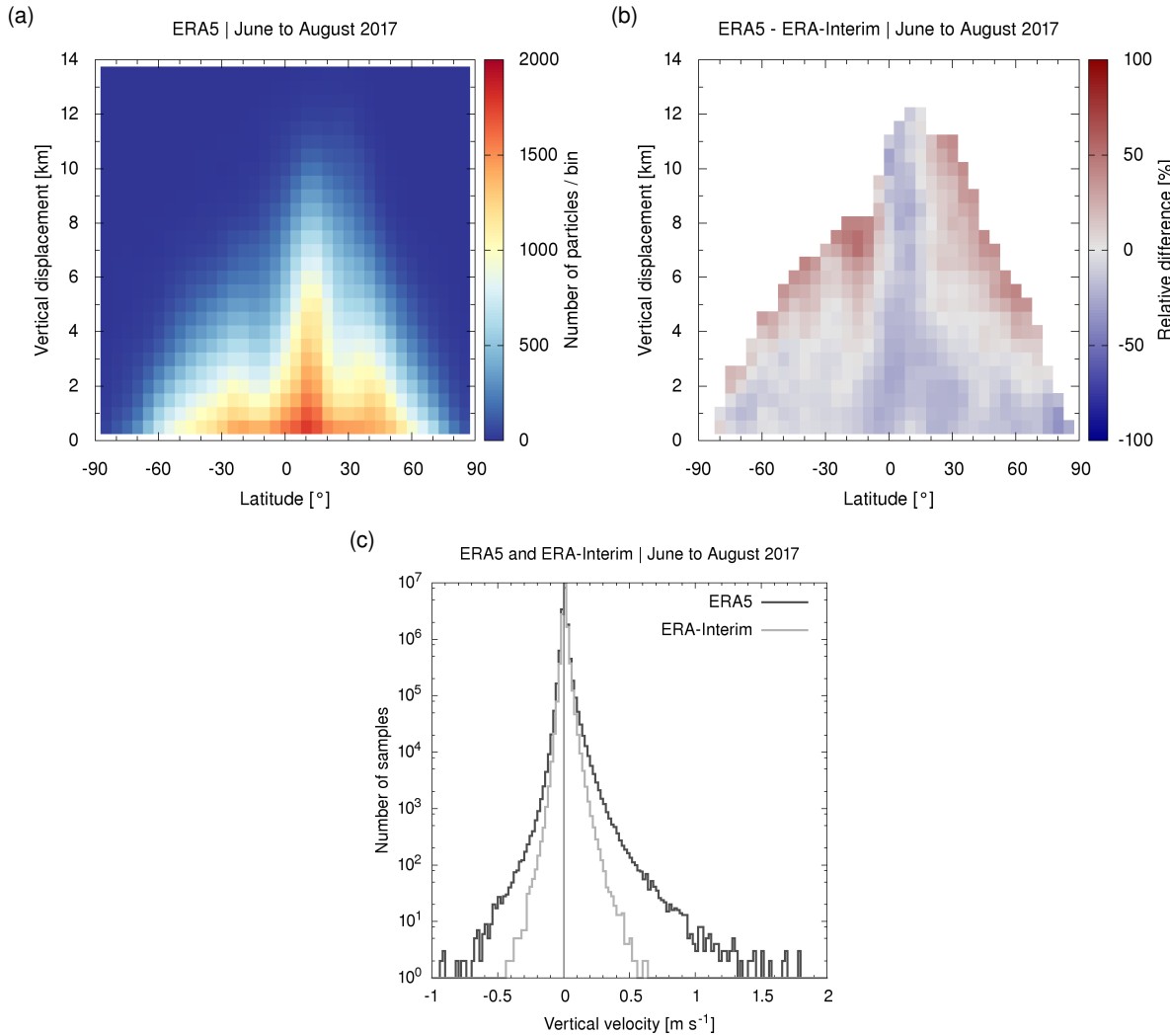

**Figure 9.** Comparison of total vertical displacements (a, b) and vertical velocities (c) of particles launched at $2-8$ km of altitude for 6 sets of ERA5 and ERA-Interim 10-day forward trajectories from June to August 2017. Only trajectories with net updraft (positive vertical displacement) after 10 days of time are considered. The bin size is $5°$ in latitude and $0.5$ km in altitude. Relative differences between ERA5 and ERA-Interim are shown only if at least 20 samples per bin are present. Vertical velocities are sampled every 6 h along the trajectories.

# ERA5 – ERA-Interim (Variability)

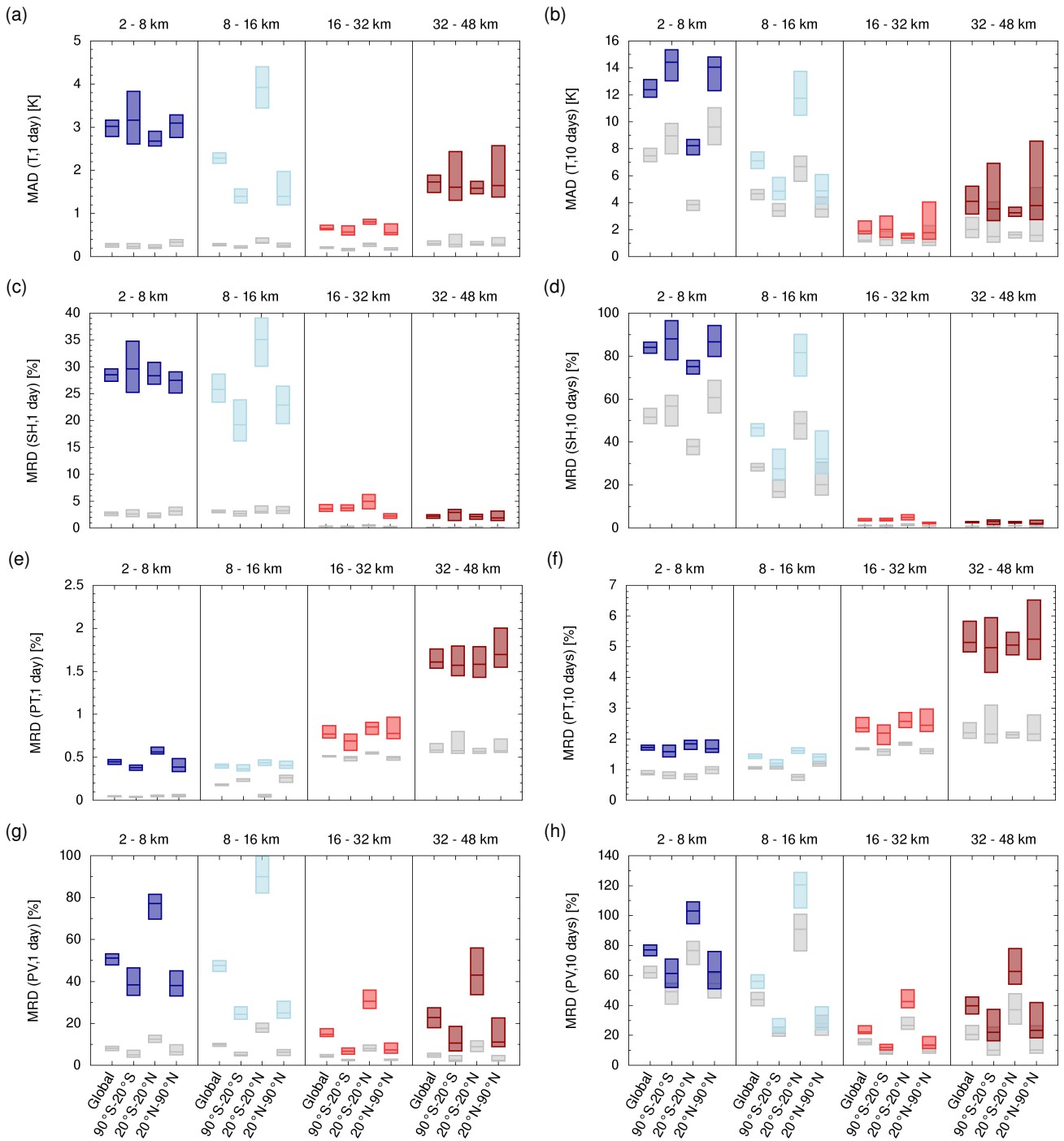

**Figure 10.** Temperature (T), specific humidity (SH), potential temperature (PT), and potential vorticity (PV) deviations between ERA-Interim and ERA5 (blue and red bars) and due to parameterized diffusion and subgrid-scale wind fluctuations (light gray bars) after 1 day (left) and 10 days (right) of time. Bars indicate the peak-to-peak range and the median of 24 trajectory simulations covering the year 2017.

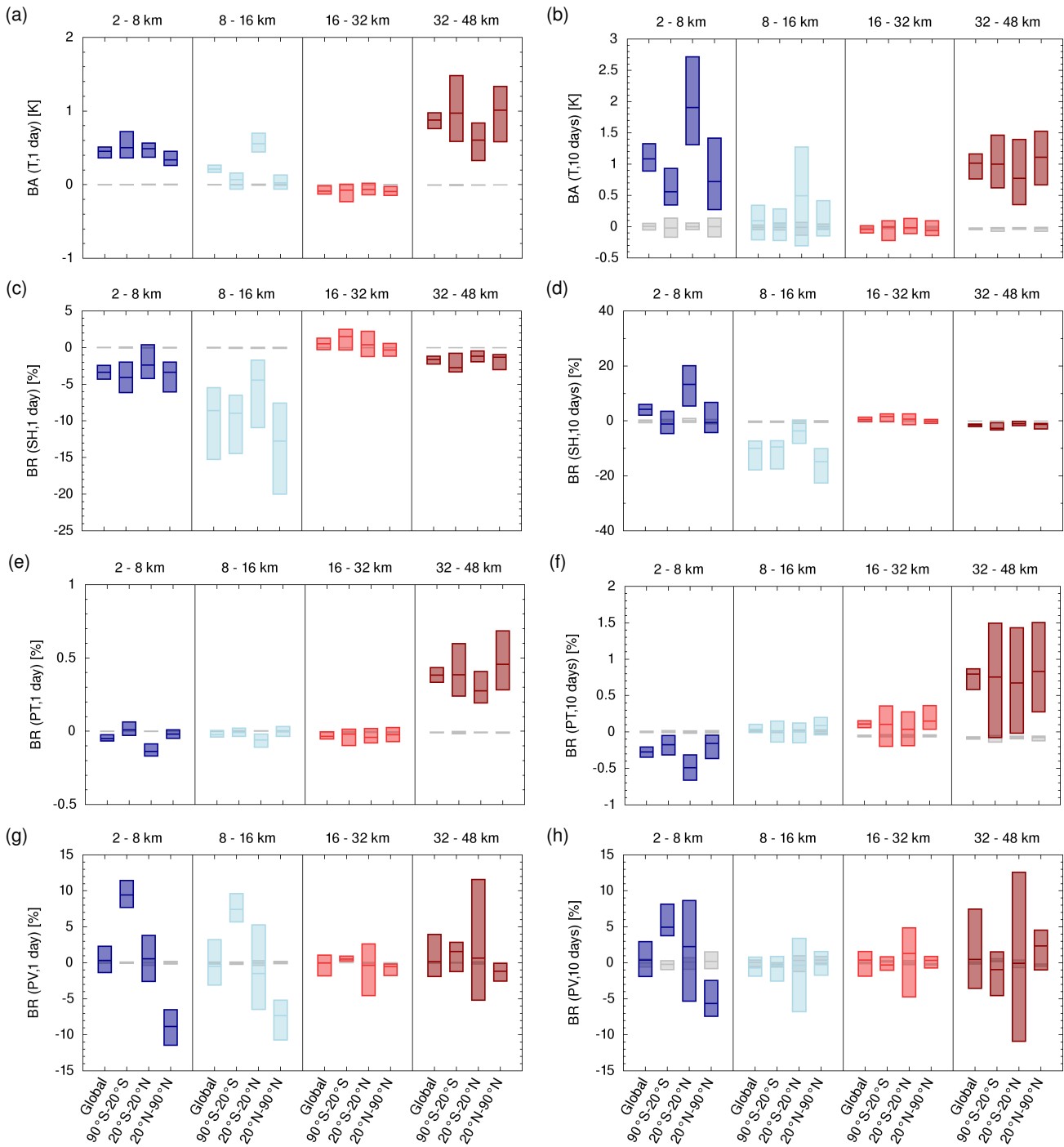

**Figure 11.** Temperature (T), specific humidity (SH), potential temperature (PT), and potential vorticity (PV) bias between ERA-Interim and ERA5 (blue and red bars) and due to parameterized diffusion and subgrid-scale wind fluctuations (light gray bars) after 1 day (left) and 10 days (right) of time. Bars indicate the peak-to-peak range and the median of 24 trajectory simulations covering the year 2017.

# Tracer conservation errors

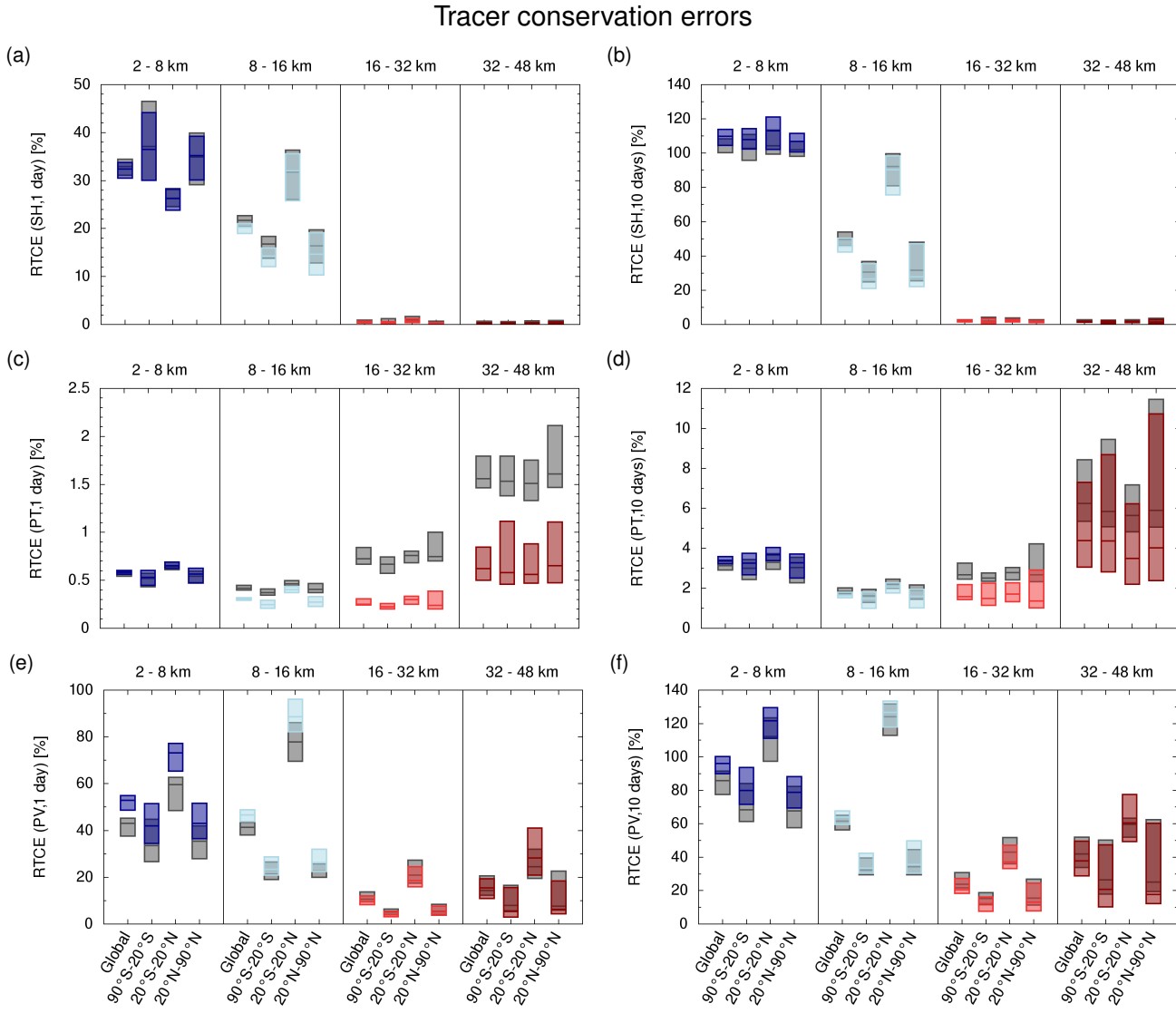

**Figure 12.** Tracer conservation errors of specific humidity (SH), potential temperature (PT), and potential vorticity (PV) in ERA5 (blue and red bars) and ERA-Interim (dark gray bars) after 1 day (left) and 10 days (right) of time. Bars indicate the peak-to-peak range and the median of 24 trajectory simulations covering the year 2017.

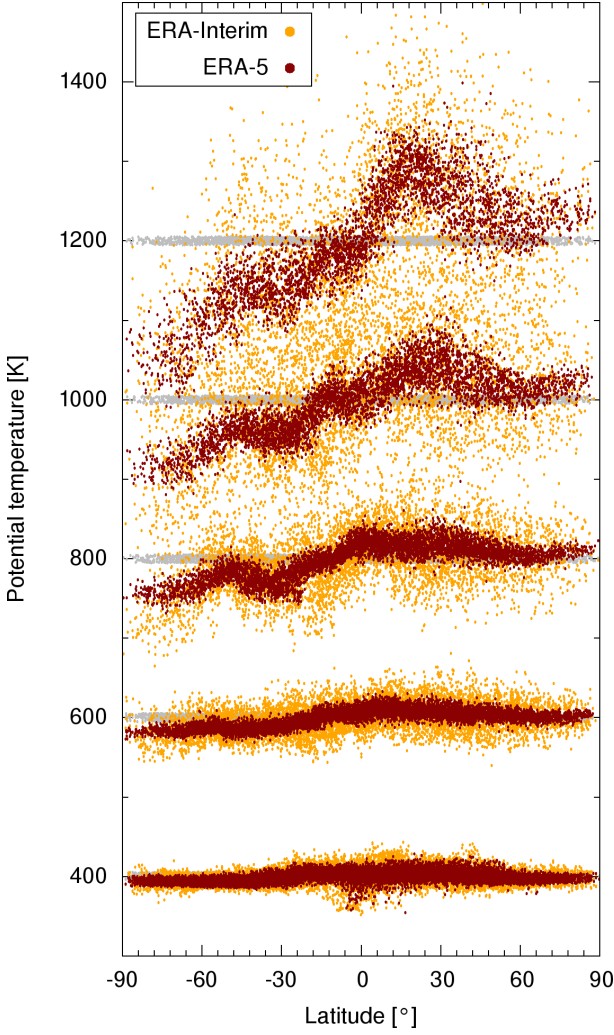

**Figure 13.** Dispersion of 10-day forward trajectories launched on 1 July 2017 at isentropic levels of 400, 600, ..., 1200 K (about 16, 24, ..., 48 km altitude; gray dots). The number of trajectory seeds varies between 12,800 at the 400 K isentropic level and 3,400 at the 1200 K isentropic level. The ERA-Interim simulations (orange dots) exhibit a larger scatter than the ERA5 simulations (red dots) after 10 days, especially at the uppermost height levels.

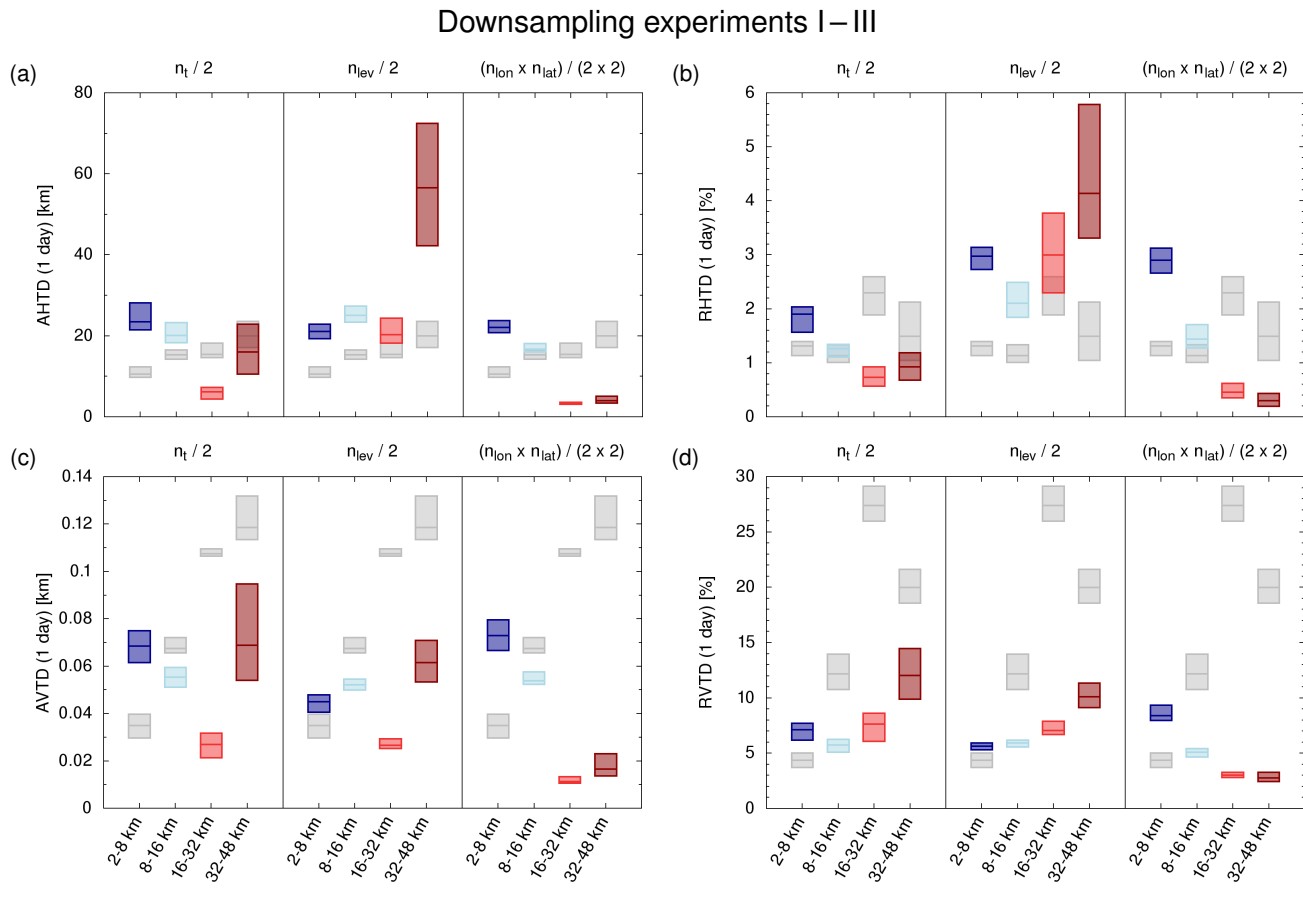

**Figure 14.** Global transport deviations after 1 day at different height levels caused by downsampling of ERA5 (blue and red bars) and due to parameterized diffusion and subgrid-scale wind fluctuations (light gray bars). The labeling of the plots refers to downsampling of the number of synoptic time steps $n_t$ (Downsampling experiment I), vertical levels $n_{lev}$ (Downsampling experiment II), and horizontal grid points $n_{lon} \times n_{lat}$ (Downsampling experiment III) of the ERA5 data, respectively.

## Downsampling experiment IV

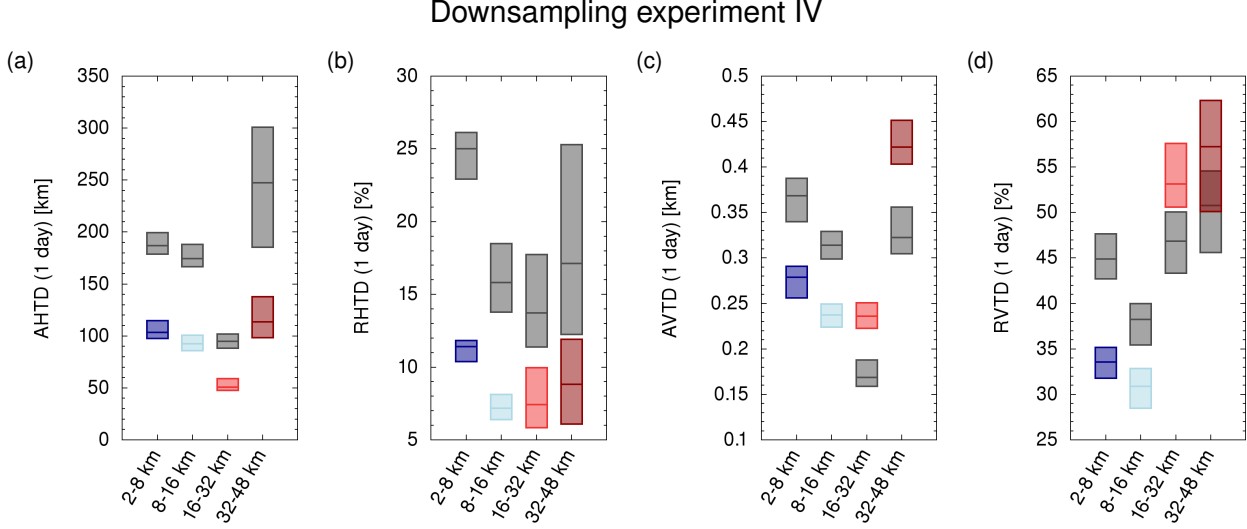

**Figure 15.** Global transport deviations of 1-day forward trajectories calculated with ERA5 data downsampled to the spatiotemporal resolution of ERA-Interim and ERA5 data at full resolution (blue and red bars). Transport deviations between ERA-Interim and ERA5 trajectories (cf. Fig. 8) are shown for reference (dark gray bars).

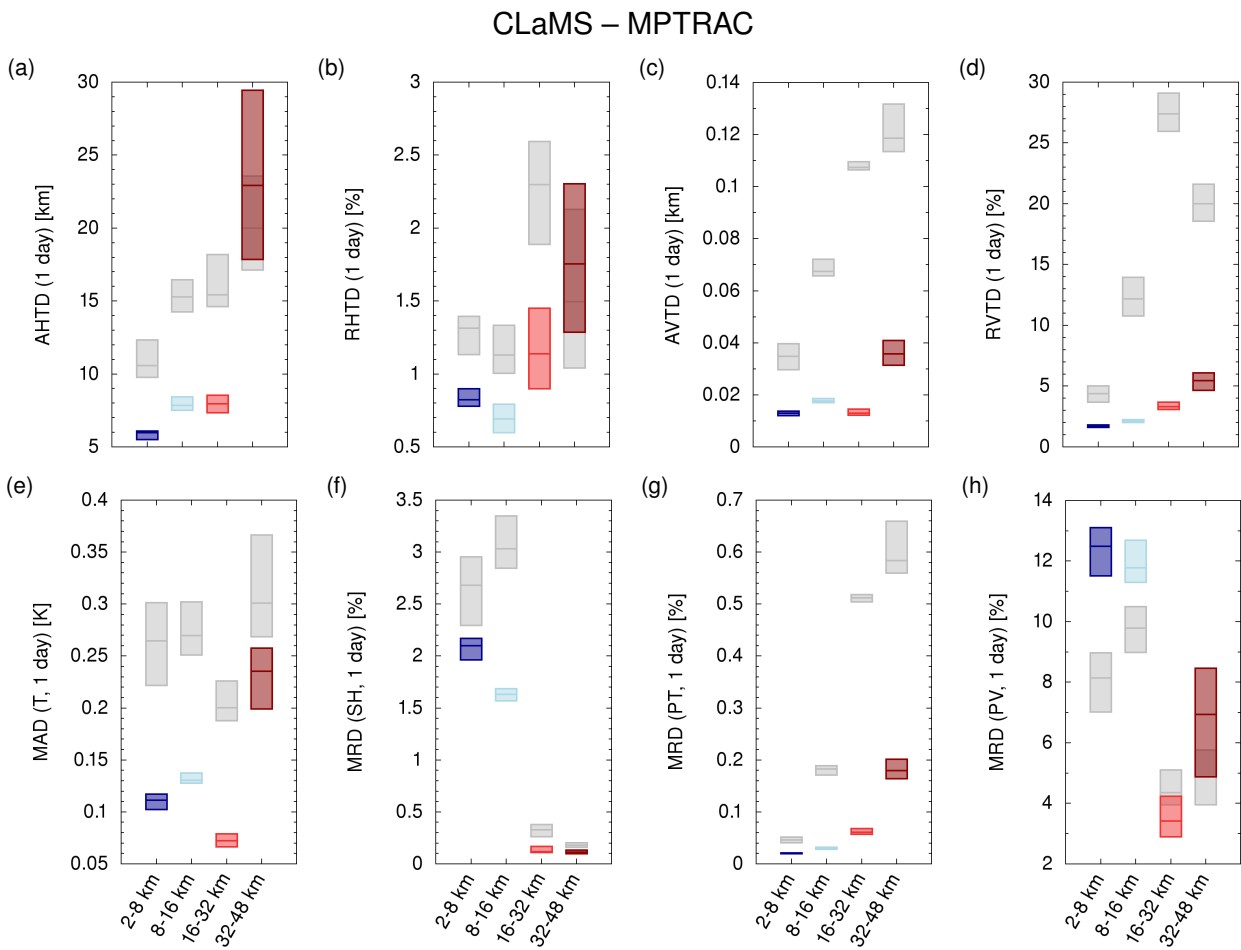

**Figure 16.** Global transport deviations (top) as well as differences in meteorological variables and dynamical tracers (bottom) of 1-day forward trajectories calculated with ERA5 data and the CLaMS or MPTRAC model (blue and red bars). Deviations due to parameterized diffusion and subgrid-scale wind fluctuations imposed on ERA5 trajectories are shown for reference (light gray bars).