# Peer review of "From ERA-Interim to ERA5: considerable impact of ECMWF's next-generation reanalysis on Lagrangian transport simulations"

_Atmospheric Chemistry and Physics, 2018_

## Referee Comment (RC1) · Anonymous Referee #1 · 4 Jan 2019

Review of *ACPD* manuscript 10.5194/acp-2018-1199 by Hoffman et al.: *From ERA-Interim to ERA5: considerable impact of ECMWF's next-generation reanalysis on Lagrangian transport simulations*

This manuscript compares atmospheric Lagrangian calculations that use two different reanalysis datasets: ERA-Interim and the new ERA5.

[Figure]

Major comments

1. §2.2.1: Please demonstrate (at least for MPTRAC) that truncation error of the Lagrangian numerical scheme is not significant. This can be done relatively easily be examining the convergence of the solver as a function of time step size. A small figure or table would suffice.

2. §2.2.2 Most of this sub-section is not of general interest and can be deleted. Some exceptions: the URLs/DOIs for the data and the size of the ERA5 data per year (indicate how many variables that represents). How much larger is a single ERA5 variable compared to ERA-Interim? Is the disk space increase for your simulations (factor of 80) due to increases in the input (winds) or output (trajectories) or both?

3. §2.3.2: I don't understand why the horizontal difference statistics are calculated approximately(?) using Cartesian coordinates when it is straightforward to calculate the actual distance for a spherical Earth using the great circle distance formula.

   The authors do not justify their use of mean absolute differences (m.a.d.) rather than root-mean-square (r.m.s.) differences, which are standard for most statistical applications, other than that Kuo et al. used m.a.d. in their 1985 paper. Later authors have simply followed Kuo et al. There are good reasons for using the standard deviation instead of the absolute deviation, as discussed in most introductory statistics books (e.g., Bulmer, 1979), and there are situations where absolute differences have advantages. It is not clear that this one of those situations.

   The choice of statistics also limits the information about differences between the two reanalyses. The authors could present valuable information about the character of those differences by showing examples of the distributions of differences

(i.e., histograms), for both horizontal and vertical dimensions. For example, are the vertical differences approximately Gaussian? Are they symmetric or skewed? Is there a bias (mean difference) as well as dispersion?

Because the horizontal differences are vectors rather than scalars, the distributions are slightly more complex, but it is easy to calculate the distance and azimuth between two particles and then create two-dimensional scatter-plots/histograms of the differences (e.g., 2-D polar plots). Again, it would be very useful to know whether there are systematic differences (biases) between the two data sets.

Similar comments apply to the statistics of the meteorological tracers.

Please define all of the symbols used in your equations.

4. §3.1: I recommend that this sub-section be discussed last in §3. The impact of 'diffusion' depends strongly on the selection of the diffusion coefficients, which is not directly related to the differences between ERA5 and ERA-I. This is discussed further in my summary recommendation.

5. §3.2: During the 10-day trajectories the particles are more likely to encounter chaotic regions in the flow where the evolution is sensitive to the initial conditions. Is that what you mean by 'separated by different airflows'?

Figure 7a (for example): If I understand this figure correctly, the horizontal differences between ERA5 and ERA-I are several hundred kilometers, while the differences due to 'diffusion' are only ∼10 km. That is the opposite of the results shown in Figure 4, where there is wide dispersion of particles by diffusion but the 'non-diffusive' trajectories are very similar. Does that mean that good agreement between the base trajectories in Figure 4 is rare? If so, why is this used as a representative sample?

How is it that the global range (min to max) is smaller than the ranges of the

individual latitude zones? Shouldn't the global range be the min and max of the all of the ranges in the latitude zones?

This figure basically provides semi-quantitative comparisons between results in different altitude layers and latitude zones. It is currently arranged to allow easiest comparisons between different altitude layers. Is that the highest priority? The figure might be easier to understand if the results were grouped by layer altitude rather than by latitude zone.

As discussed above, it is important to know whether these differences are systematic or random.

6. §3.3 and §3.4: How much of the difference between the 'tracer' variables is due to differences in the *trajectories* and how much is due to differences in the *analyses*? If you use an ERA5 trajectory to evaluate a variable in the ERA5 analysis *and* in the ERA-I analyses, how large are the differences?

7. §3.5: Users may choose to simply downsample the ERA5 data in space and time rather than filtering the data first. How does downsampling the data compare to low-pass filtering and then downsampling? Are the errors comparable, or does the filtering significantly improve the results?

Minor comments

1. The authors should cite the seminal papers on Lagrangian atmospheric methods, not just their own recent papers. For example, Djuric, *J Meteorology*, 1961 and the papers cited therein lay out many of the same issues discussed here. More recent examples include Hsu (*JAS*, 1980) and Kida (*J Met Soc Japan*, 1983).

2. Table 1: Please provide the dimensions of the horizontal transform grids (e.g., 512 ×256 for ERA-I) in addition to the spatial resolution. If there are standard

grids on which the analyses are made available (e.g., regularly-spaced grids that differ from the models' spectral transform grids), please include that information also.

3. Paragraph beginning on page 2 line 2: Lagrangian models may or may not include parameterizations for 'diffusion'. Whether they do or not, I think it is important to maintain the distinction between *molecular diffusion* (e.g., random walks of molecules due to collisions) and the *stirring* of fluid by unresolved scales of motion, which is often represented as a diffusive process by using an eddy diffusivity coefficient. Because molecular diffusion acts on very small spatial scales, in the atmosphere it is almost always insignificant compared to unresolved scales of motion, whether turbulent or not.

   Lagrangian calculations do not necessarily involve a grid (regular or irregular). Particles can be initialized randomly within the domain, for example.

   Lagrangian calculations are not *explicitly* affected by numerical diffusion, but the wind fields used for Lagrangian calculations are almost always produced by a model or data assimilation system. The model winds *are* influenced by both the explicit and numerical diffusion in the model.

   The real power of Lagrangian methods is that, given a *continuous* wind field, which is typically provided by a combination of velocities at discrete grid points and a space-time interpolation scheme, a Lagrangian solver can compute the trajectory of a particle with essentially arbitrary precision. The interpolated wind field is an approximation of the real wind field, of course, and is typically designed to be smooth at the grid scale (i.e., at least piece-wise continuous). Lagrangian methods can avoid the diffusion of passive tracers that is always present to some degree in an Eulerian model and provide an effectively higher spatial resolution.

4. Equation 9: Strictly speaking these are not errors because the quantities in question are not perfectly conserved. Some degree of non-conservation is to be ex-
pected. You are really evaluating the non-conservation, which may be real, due to diabatic heating or dissipation, or a result of various sources of error.

Instead of 'dynamical tracer' you might say 'quasi-conserved variable'.

5. Figure 4: Is the horizontal dispersion in Figure 4 largely due to a combination of vertical displacement by the sub-gridscale wind parameterization and vertical shear of the resolved horizontal wind?

6. Page 10 line 22: If the horizontal displacement differences are several thousand kilometers, differences in the planetary vorticity $f$ may also contribute to the differences in PV. I think it should be straightforward to sort out the contributions of $\zeta$, $f$, and stability.

7. Page 16 line 1: The stratosphere is not 'dynamically less disturbed than the troposphere', but the spectrum of motion is redder in both space and time, so the actual flow is likely to be better represented on the model grid than in the troposphere.

Recommendation

This will be a very useful paper for both research and applications of Lagrangian methods in the atmosphere. I recommend publication after some revisions. My greatest concern is discussed in major comment #3. I think it would be of great benefit to see some actual distributions of differences between the two reanalyses, rather than only descriptive statistics. Also, the authors should explain why the absolute deviations are more useful for this application than standard deviations.

My second major concern is about the comparisons between calculations with and without 'diffusion'. A serious problem throughout the paper is that you don't really know the magnitude of the dispersion by sub-gridscale components of the flow. You only

know the impact of *parameterized* sub-gridscale winds, which may be very different from reality. You *can* argue that, at least with current estimates of the magnitude of the sub-gridscale dispersion, its impact is smaller than the differences resulting from differences at resolved scales.

In general I feel the paper attempts to do to much and presents too many results with too much detail (especially figures 7, 9, 10, 12, and 13). The results are comprehensive rather than illustrative, which makes it quite difficult for the reader to synthesize the results and extract the essential points. Perhaps the complete results could go into appendices or supplementary materials (as tables?), with only the most significant results discussed in the main text.

---

## Referee Comment (RC2) · Anonymous Referee #2 · 10 Jan 2019

This is a well-written, well-designed, and well-structured manuscript. The authors have performed a careful analysis of transport characteristics and their differences between ECMWF's ERA-Interim and ERA-5 reanalyses. I have only a few suggestions and corrections for the authors.

Page 9, lines 7-8: suggest revising "should be conserved" to "can be conserved" or "are often mostly conserved". As acknowledged later in the manuscript, conservation of PV and other fields relies on a set of assumed conditions or histories of the tracked air parcels, which are rarely met in the troposphere and often not met in the UTLS.

Page 9, line 29: delete extra "the"

[Figure]

Page 12, lines 31-32: "...by means of the mean relative..." is poor phrasing. Suggest revising to "...using the mean relative..."

Section 3.4: Suggest adding text to the third and fourth paragraphs outlining when potential temperature and PV are not conserved, as was done for specific humidity here.

Page 18, lines 2-3: assuming my interpretation is correct, suggest clarifying "(e.g., by means of ensemble simulations)" to "(e.g., by means of ensemble trajectory simulations)". If not, please clarify as appropriate.

Figure 1: for the horizontal axis, I would suggest using "layer depth" instead of "layer width" given the focus on the vertical dimension.

Figure 5: The labels for each panel here are confusing. They say "w/o diffusion", but what is being shown are deviations due to diffusion and subgrid-scale mixing. Please revise/simplify these labels and the caption to avoid confusion. This is also true for Figure 6!

Figure 6: In addition to the label issue, how are these particles collected for analysis? Based on initial position, final position, some other way? Its not clear how these regional analyses are done. A simple clarification in the caption should suffice.

Figure 12: I suggest labeling each panel by experiment number to improve reader evaluation.

---

## Referee Comment (RC3) · Anonymous Referee #3 · 1 Feb 2019

This work is a very useful comparison of the transport properties of the new ECMWF reanalysis ERA5 compared with the ERA-I and should help in popularizing the usage of ERA5. There are, however a few points that need improvements or clarifications as listed below

1) I am unsure that the description of meteorological condition during 2017 is relevant to the scope of this work. This subsection and the two figures 2 and 3 are not exploited in the sequel and are only distracting.

2) I am also unsure that the details of disk storage at FZJ in 2.2.2 are of public interest if the ERA5 deposit is only for internal usage.

[Figure]

3) The dispersion of parcels by diffusion is effective only over a couple of days. The subsequent dispersion is due to the explicit wind shear and strain of the resolved eddies. This needs to be clarified with the help of figure 5. Actually, it seems that we see several regimes in figure 5: a diffusive regime with a sqrt(t) behavior (that generates diverging RVTD at small t), an exponential regime characteristic of chaotic dispersion and a linear regime due to large eddy dispersion.

4) The very good conservation of the potential temperature in the ERA5 compared to ERA-I is an interesting and somewhat puzzling point. It could be due to the improvement in the transport in the model or to the fact that the troncature of the model is more consistent with the spectrum of motion and that rejected modes that generate aliases are only weakly excited. Another possibility is that the temperature assimilation increments are much reduced in ERA5 with respect to ERA-I. In any case, this circumstance should facilitate the determination of appropriate vertical diffusivity to represent the lacking subgrid-scale motion in Lagrangian trajectories. It is quite possible that the required value should be smaller than 0.1 m2 s-1 found in previous studies based on ERA-40 winds.

5) It is important to mention that convective properties are quite different in the ERA5 which displays much more intermittency than the ERA-Interim.

6) I do not think that the layer 8-16 km corresponds to the UT/LS in the tropics as it encompasses the mean layer of convective detrainment at 12-13 km.

7) I do not see what additional information is brought by figure 11.

8) The comment about assimilation increments of the vertical velocities must be removed or rewritten since there is no assimilation increment of the vertical velocities for hydrostatic models where the vertical velocity is a diagnosed quantity.

9) The downsampling procedure does not really leads to a fair comparison between ERA5 and ERA-I. The spatial downscaling can be seen as a smoothing but the temporal downscaling cannot since the archived wind data are not time integrated quantities but instantaneous values. This section is quite important as users might not all be able to download and store hourly ERA5 data at maximum spatial resolution.

---

## Author Comment (AC1) · 25 Feb 2019

**Reply to review comments**

We thank the reviewers for the time and effort spent on the manuscript and for providing helpful comments. We considered all comments and hope that the revised draft properly addresses the open issues. Please find our point-by-point replies below (colored in blue). Note that references to figures and tables in this reply refer to the ACPD version of our paper. A revised manuscript with tracked changes has been attached to this document.

**Reviewer #1**

This manuscript compares atmospheric Lagrangian calculations that use two different re-analysis datasets: ERA-Interim and the new ERA5.

Major comments

1. §2.2.1: Please demonstrate (at least for MPTRAC) that truncation error of the Lagrangian numerical scheme is not significant. This can be done relatively easily be examining the convergence of the solver as a function of time step size. A small figure or table would suffice.

A detailed analysis of the truncation errors of the MPTRAC trajectory calculations was presented by Rößler et al. (2018). However, as an example, Fig. 1 in this reply illustrates the convergence of the simulation results for ERA5 data. It shows the dependence of the transport deviations after 1 day and 10 days on the time step $\Delta t$ of the numerical solver (mid-point scheme) used by MPTRAC. Simulations with $\Delta t = 60\,\text{s}$ were selected for reference. The transport deviations for $\Delta t = 240\,\text{s}$ (choice for the simulations in the paper) are much smaller than any others found in the paper, which demonstrates that truncation errors do not contribute significantly to the results. The same analysis was conducted for ERA-Interim, showing that a time step $\Delta t = 600\,\text{s}$ is suitable for that data set. For the CLaMS model the convergence has not been tested here. However, CLaMS applies a higher-order numerical integration scheme than MPTRAC and we used the same time steps for CLaMS and MPTRAC, so we expect comparable or even better accuracy for CLaMS. In Sect. 2.2.1 we added "Sensitivity tests showed that the time steps are small enough so that truncation errors do not contribute significantly to the simulation results."

2. §2.2.2 Most of this sub-section is not of general interest and can be deleted. Some exceptions: the URLs/DOIs for the data and the size of the ERA5 data per year (indicate how many variables that represents). How much larger is a single ERA5 variable compared to ERA-Interim? Is the disk space increase for your simulations (factor of 80) due to increases in the input (winds) or output (trajectories) or both?

We agree that the material presented in Sect. 2.2.2 is mostly of a technical nature and may only be interesting for a smaller number of readers. Therefore, we made this section

[Figure]

Figure 1: Transport deviations as a function of the time step $\Delta t$ of the numerical solver used by MPTRAC to calculate ERA5 trajectories.

an appendix of the paper, so that the main text becomes more focused and concise. In the main text we retained the references to the data sets, moved the information regarding the size of the variables for ERA5 and ERA-Interim to the introduction, added the number of variables to Table 1, and clarified that the factor of 80 refers to input data.

3. §2.3.2: I don't understand why the horizontal difference statistics are calculated approximately(?) using Cartesian coordinates when it is straightforward to calculate the actual distance for a spherical Earth using the great circle distance formula.

To clarify, we added "Euclidean distances approximate great circle distances with good accuracy ($\geq 97\%$ up to $5000\,\mathrm{km}$ of distance)." Here, the value of $5000\,\mathrm{km}$ covers the largest distances analyzed in the paper (Fig. 7b). Most AHTDs considered here are much smaller and the approximation will be even better.

The authors do not justify their use of mean absolute differences (m.a.d.) rather than root-mean-square (r.m.s.) differences, which are standard for most statistical applications, other than that Kuo et al. used m.a.d. in their 1985 paper. Later authors have simply followed Kuo et al. There are good reasons for using the standard deviation instead of the absolute deviation, as discussed in most introductory statistics books (e.g., Bulmer, 1979), and there are situations where absolute differences have advantages. It is not clear that this one of those situations.

Both, mean absolute deviation (MAD) and standard deviation (SD) are common measures of variability. While SD has distinct mathematical advantages, in particular for Gaussian distributions and statistical tests, MAD is more "intuitive" (a philosophical argument), but also more robust against outliers and for non-Gaussian distributions (a practical argument). The distribution of horizontal distances underlying the calculation of AHTDs is non-Gaussian, because it is calculated only from non-negative values. Typically, the distributions of AHTDs are skewed towards large outliers (e. g., Stohl et al., 2001; Rößler et al., 2018). This can also be seen from the trajectory example shown in Fig. 4 of the paper. Fig. 2 in this reply provides corresponding histograms of the distributions of particle positions and meteorological variables for further illustration. This may provide good reason to use MADs instead of SDs for transport deviations and explain why many other researchers followed Kuo et al. (1985) and used MADs rather than SDs for Lagrangian transport analyses (e. g., Rolph and Draxler, 1990; Stohl et al., 1995; Stohl and Seibert, 1998; Stohl, 1998; Stohl et al., 2001; Harris et al., 2005; Riddle et al., 2006; Miltenberger et al., 2013, and references therein).

The choice of statistics also limits the information about differences between the two reanalyses. The authors could present valuable information about the character of those differences by showing examples of the distributions of differences (i.e., histograms), for both horizontal and vertical dimensions. For example, are the vertical differences approximately Gaussian? Are they symmetric or skewed? Is there a bias (mean difference) as well as dispersion?

As discussed above, we generally do not expect the distributions of the particle positions

[Figure]

Figure 2: Histograms of particle positions and meteorological variables for the stratospheric trajectory example shown in Fig. 4 of the paper.

as well as their deviations to be Gaussian, in particular after longer time periods such as 5 – 10 days. At that time particles often have entered chaotic regions and have been dispersed by divergent flows. This behavior is illustrated by Fig. 2 in this reply, but has also be noted in other studies (e. g. Stohl, 1998; Stohl et al., 2001; Rößler et al., 2018). For this reason, we think that transport deviations (AHTDs, AVTDs, RHTDs, RVTDs) are the most appropriate measures.

Because the horizontal differences are vectors rather than scalars, the distributions are slightly more complex, but it is easy to calculate the distance and azimuth between two particles and then create two-dimensional scatter-plots/histograms of the differences (e.g., 2-D polar plots). Again, it would be very useful to know whether there are systematic differences (biases) between the two data sets.

A more detailed analysis of trajectory deviations in terms of their along-track and directional differences might be interesting, but we are afraid that this is beyond the scope of our current study. The results of a more detailed analysis may probably depend strongly on the individual wind fields in the troposphere and stratosphere. Therefore, such an analysis should perhaps be focused more closely on individual phenomena (e. g., convection, jets, etc.). We will consider this for future work.

Similar comments apply to the statistics of the meteorological tracers.

We decided for MADs rather than SDs to analyze the tracer statics for consistency with the analysis of transport deviations. Considering that the distributions of the transport deviations of the trajectories can be skewed and affected by outliers, this will also apply for the corresponding tracer distributions along the trajectories. Therefore, MADs are likely more appropriate than SDs to analyze these distributions. The use of MADs and MRDs facilitates comparison with the results of Stohl and Seibert (1998).

In the manuscript, we added the following explanation: "Here, we chose MADs rather than standard deviations for the statistical analysis to achieve consistency with the definitions of the transport deviations (AHTDs and AVTDs). Also, MADs are more robust than standard deviations against outliers. For a more detailed discussion of the advantages and disadvantages of using MADs versus standard deviations see Willmott and Matsuura (2005) and Chai and Draxler (2014)." We replaced the reference to Stohl et al. (1995) by an earlier paper (Rolph and Draxler, 1990), and we added a reference to Stohl (1998), as it provides a more detailed discussion regarding the calculation of transport deviations. The papers of Willmott and Matsuura (2005) and Chai and Draxler (2014) discuss the issue of MADs versus SDs more generally.

Please define all of the symbols used in your equations.

We checked that all symbols have been defined.

4. §3.1: I recommend that this sub-section be discussed last in §3. The impact of 'diffusion' depends strongly on the selection of the diffusion coefficients, which is not directly related to the differences between ERA5 and ERA-I. This is discussed further in my

summary recommendation.

We considered moving Sect. 3.1 to the end of Sect. 3 or turn it into an appendix, but we concluded that despite some difficulties regarding the interpretation of the simulated diffusion, we should keep it in place as it introduces a number of aspects. In addition to the case study (Fig. 4), Sect. 3.1 introduces the statistical analysis of the transport deviations (by means of Figs. 5 and 6), which we think should be presented before the summary plots (Figs. 7, 9, 10 and 12 – 14) are shown. In the revised manuscript we tried to better clarify our approach of comparing the differences between ERA5 and ERA-Interim to simulated diffusion. Please see reply to summary recommendation.

5. §3.2: During the 10-day trajectories the particles are more likely to encounter chaotic regions in the flow where the evolution is sensitive to the initial conditions. Is that what you mean by 'separated by different airflows'?

We rephrased this to "The 10-day transport deviations are typically strongly affected by the individual atmospheric conditions, e.g., as particles enter chaotic regions and are dispersed by divergent flows.".

Figure 7a (for example): If I understand this figure correctly, the horizontal differences between ERA5 and ERA-I are several hundred kilometers, while the differences due to 'diffusion' are only 10 km. That is the opposite of the results shown in Figure 4, where there is wide dispersion of particles by diffusion but the 'non-diffusive' trajectories are very similar. Does that mean that good agreement between the base trajectories in Figure 4 is rare? If so, why is this used as a representative sample?

We introduced Figure 4 as an "illustrative" rather than as a "representative" example. The figure illustrates how diffusion and subgrid-scale wind fluctuations can influence trajectory calculations in the model and how these simulations can help to judge if differences between test and reference trajectories are significant or not. In the revised manuscript, we clarified that there is "particularly good agreement" (instead of "rather good agreement") between the ERA5 and ERA-Interim base trajectories in this case.

How is it that the global range (min to max) is smaller than the ranges of the individual latitude zones? Shouldn't the global range be the min and max of the all of the ranges in the latitude zones?

This is not expected because the peak-to-peak ranges cover the ranges of the *mean* transport deviations of the individual 10-day calculations. Larger transport deviations can be present in specific latitude bands, but these larger deviations are suppressed in global means because of averaging over all latitude bands.

This figure basically provides semi-quantitative comparisons between results in different altitude layers and latitude zones. It is currently arranged to allow easiest comparisons between different altitude layers. Is that the highest priority? The figure might be easier to understand if the results were grouped by layer altitude rather than by latitude zone.

We tested this rearrangement and found it makes the figure much easier to understand. Thank you for this helpful suggestion! We also adapted Figs. 9 and 10 accordingly.

As discussed above, it is important to know whether these differences are systematic or random.

We agree that a more comprehensive characterization of the tracer differences can be helpful and analyzed the biases between the ERA5 and ERA-Interim tracer distributions. Overall, we found that the biases are notably smaller than the variability as measured by the MADs or MRDs. Nevertheless, these additional results seem worthwhile reporting in the revised paper. We added the equations used to calculate the absolute and relative bias in Sect. 2.3.2 and the results to Sect. 3.3.3 (including a new Fig. 10).

6. §3.3 and §3.4: How much of the difference between the 'tracer' variables is due to differences in the trajectories and how much is due to differences in the analyses? If you use an ERA5 trajectory to evaluate a variable in the ERA5 analysis and in the ERA-I analyses, how large are the differences?

Our analysis aims to quantify the accumulated effect or 'total error' due to the transport deviations between the trajectories and due to the tracer differences. Mixing ERA5 and ERA-Interim data in the Lagrangian transport analysis would introduce inconsistencies between the forecast models prognostic variables (wind, temperature, specific humidity) and the dynamical tracers (potential temperature, potential vorticity), which we think should be avoided. Direct differences between ERA5 and ERA-Interim tracer data can be estimated without Lagrangian transport simulations. As the trajectory positions provide nearly homogeneous coverage of the troposphere and stratosphere in our case, we can simply calculate the differences of the ERA5 and ERA-Interim fields on their regular grids at synoptic timesteps to get the deviations between the tracer data. This was already done for temperature and zonal winds in Fig. 3 of the manuscript. For reference in other section of the revised manuscript, we extended this analysis to also cover specific humidity. We also added a new Fig. 4, so that Figs. 3 and 4 provide more information on direct biases between ERA5 and ERA-Interim data in January and July 2017, respectively. The discussion in Sect. 2.1.2 was revised accordingly.

7. §3.5: Users may choose to simply downsample the ERA5 data in space and time rather than filtering the data first. How does downsampling the data compare to low-pass filtering and then downsampling? Are the errors comparable, or does the filtering significantly improve the results?

Figure 3 in this reply shows the results of a sensitivity test for downsampling experiment IV, in which we skipped the low-pass filtering completely before downsampling. Compared with the original results shown in Fig. 13 of the paper, this test shows that low-pass filtering may significantly improve the results, in particular for vertical transport deviations in the stratosphere. Please note that we also had to revise Sect. 3.5 following comments provided by Reviewer #3.

[Figure]

Figure 3: Results of a sensitivity test for downsampling experiment IV, with low-pass filtering being switched off.

Minor comments

1. The authors should cite the seminal papers on Lagrangian atmospheric methods, not just their own recent papers. For example, Djuric, J Meteorology, 1961 and the papers cited therein lay out many of the same issues discussed here. More recent examples include Hsu (JAS, 1980) and Kida (J Met Soc Japan, 1983).

We added these references.

2. Table 1: Please provide the dimensions of the horizontal transform grids (e.g., 512 x 256 for ERA-I) in addition to the spatial resolution. If there are standard grids on which the analyses are made available (e.g., regularly-spaced grids that differ from the models' spectral transform grids), please include that information also.

We added this information to Table 1. Standard grids have not been defined. The user of ECMWF's MARS archives chooses the transform grid when retrieving the data.

3. Paragraph beginning on page 2 line 2: Lagrangian models may or may not include parameterizations for 'diffusion'. Whether they do or not, I think it is important to maintain the distinction between molecular diffusion (e.g., random walks of molecules due to collisions) and the stirring of fluid by unresolved scales of motion, which is often represented as a diffusive process by using an eddy diffusivity coefficient. Because molecular diffusion acts on very small spatial scales, in the atmosphere it is almost always insignificant compared to unresolved scales of motion, whether turbulent or not.

We deleted the phrase "(i. e., advection and diffusion)" to avoid any confusion or a more detailed discussion in this introductory paragraph.

Lagrangian calculations do not necessarily involve a grid (regular or irregular). Particles can be initialized randomly within the domain, for example.

We deleted the phrase saying "The particles are distributed on an irregular grid [...]" to avoid any confusion.

Lagrangian calculations are not explicitly affected by numerical diffusion, but the wind fields used for Lagrangian calculations are almost always produced by a model or data assimilation system. The model winds are influenced by both the explicit and numerical diffusion in the model.

We rephrased the corresponding sentences to "A major advantage is that the spatial resolution of Lagrangian transport simulations is not limited to a regular grid. The approach can avoid the numerical diffusion of passive tracers that is always present to some degree in Eulerian models. The method is therefore well capable of representing small-scale features such as filaments of tracers associated with long-range transport."

The real power of Lagrangian methods is that, given a continuous wind field, which is typically provided by a combination of velocities at discrete grid points and a space-time interpolation scheme, a Lagrangian solver can compute the trajectory of a particle with essentially arbitrary precision. The interpolated wind field is an approximation of the real wind field, of course, and is typically designed to be smooth at the grid scale (i.e., at least piece-wise continuous). Lagrangian methods can avoid the diffusion of passive tracers that is always present to some degree in an Eulerian model and provide an effectively higher spatial resolution.

Please see reply to previous comment.

4. Equation 9: Strictly speaking these are not errors because the quantities in question are not perfectly conserved. Some degree of non-conservation is to be expected. You are really evaluating the non-conservation, which may be real, due to diabatic heating or dissipation, or a result of various sources of error.

We clarified that the dynamical tracers "can be conserved" (following a comment of Reviewer #2) and further added "Note that in reality part of the RTCE is due to non-conservation, e. g., due to diabatic heating or dissipation."

Instead of 'dynamical tracer' you might say 'quasi-conserved variable'.

Following Andrews et al. (1987), we used the term "dynamical tracer" (i. e., potential temperature and potential vorticity) to distinguish it from "chemical tracers" and other meteorological variables (i. e., specific humidity and temperature). We retained the term "dynamical tracer" for now, as additional clarification was provided regarding its meaning.

5. Figure 4: Is the horizontal dispersion in Figure 4 largely due to a combination of vertical displacement by the sub-gridscale wind parameterization and vertical shear of the resolved horizontal wind?

Figure 4 in this reply shows the zonal mean and standard deviations of the ERA5 zonal winds in January 2017. The region of the example trajectory (about $20-21$ km of altitude, $25-40°$N) is located slightly below and to the south of the polar jet. Some vertical shear

of the zonal winds may be expected due to meandering of the jet.

[Figure]

Figure 4: Zonal mean and standard deviation of ERA5 zonal winds in January 2017.

We conducted additional simulations in which individual horizontal and vertical components of the parameterization of diffusion and sub-grid scale wind fluctuations of MPTRAC have been switched on or off. We found that the diffusion of the particles is mostly determined by vertical diffusivity in our example. A more detailed analysis shows that the diffusivity arises from using a fixed vertical diffusivity of $D_z = 0.1\,\mathrm{m}^2/\mathrm{s}$ in the stratosphere rather than from downscaling of the vertical wind components (see Fig. 5 in this reply). This finding is consistent with Fig. 5 of the paper, showing a distinct scaling behavior of AVTD $\propto \sqrt{t}$ in the stratosphere.

We added the following explanation to the paper: "A more detailed analysis showed that the dispersion of the ERA5 trajectory set seen in this particular example is mostly due to a combination of vertical displacements due to the use of a constant vertical diffusivity $D_z = 0.1\,\mathrm{m}^2/\mathrm{s}$ in the stratosphere (Legras et al., 2003; Stohl et al., 2005) and vertical shear of the resolved horizontal winds. Note that the resulting horizontal and vertical distributions of the particle positions became non-Gaussian."

6. Page 10 line 22: If the horizontal displacement differences are several thousand kilometers, differences in the planetary vorticity f may also contribute to the differences in PV. I think it should be straightforward to sort out the contributions of $\zeta$, f, and stability.

Unfortunately, it is not easy for us to sort out the individual contributions to potential vorticity as we are calculating it on pressure levels rather than on isentropic surfaces. Following Eliassen (1987),

$$Q = g\left[(f + \zeta_p)\left(-\frac{\partial \theta}{\partial p}\right) + \frac{\partial v}{\partial p}\left(\frac{\partial \theta}{\partial x}\right)_p - \frac{\partial u}{\partial p}\left(\frac{\partial \theta}{\partial y}\right)_p\right],$$

[Figure]

Figure 5: Sensitivity tests regarding influence of sub-grid scale wind fluctuations on the stratospheric trajectory example discussed in the paper (Fig. 4 and Sect. 3.1). Shown are maps of trajectory positions based on original data including all parameterizations (top, left), a simulation with only a constant vertical diffusivity $D_z = 0.1\,\mathrm{m}^2/\mathrm{s}$ being considered (top, right), only horizontal components of the parameterization being considered (bottom, left), and only vertical components being considered (bottom, right).

$$\zeta_p = \left(\frac{\partial v}{\partial x}\right)_p - \frac{1}{\cos\varphi}\left[\frac{\partial(u\cos\varphi)}{\partial y}\right]_p .$$

We agree that significant variability can be present in absolute vorticity $f$, so we rephrased "As potential temperature is nearly constant in this case, the variability in potential vorticity is due to variability in relative vorticity as calculated from the horizontal winds and variability in absolute vorticity due to the particles being dispersed to different latitudes."

7. Page 16 line 1: The stratosphere is not 'dynamically less disturbed than the troposphere', but the spectrum of motion is redder in both space and time, so the actual flow is likely to be better represented on the model grid than in the troposphere.

We rephrased "This may reflect the reduced sensitivity of the stratosphere to downsampling in the horizontal direction and in time, as the stratosphere is dynamically more stable and has a redder spectrum of motion than the troposphere."

Recommendation

This will be a very useful paper for both research and applications of Lagrangian methods in the atmosphere. I recommend publication after some revisions. My greatest concern is discussed in major comment #3. I think it would be of great benefit to see some actual distributions of differences between the two reanalyses, rather than only descriptive statistics. Also, the authors should explain why the absolute deviations are more useful for this application than standard deviations.

We thank you for the supporting statements! In this reply and in the revised manuscript we tried to better motivate our choice of the specific set of descriptive statistics selected for the analysis. We decided to not include additional figures showing histograms of the distributions of particle positions and tracer distributions in the revised manuscript, but extended the explanations and discussion, accordingly.

My second major concern is about the comparisons between calculations with and without 'diffusion'. A serious problem throughout the paper is that you don't really know the magnitude of the dispersion by sub-gridscale components of the flow. You only know the impact of parameterized sub-gridscale winds, which may be very different from reality. You can argue that, at least with current estimates of the magnitude of the sub-gridscale dispersion, its impact is smaller than the differences resulting from differences at resolved scales.

We agree that the parameterization of 'diffusion' is a difficult problem for Lagrangian transport models and that diffusion and subgrid-scale winds parameterizations bear some uncertainty. However, all main results regarding the differences of the ERA5 and ERA-Interim data sets are assessed independently and without simulating diffusion. The uncertainties due to diffusion are presented here simply as guiding values, to help judge if other transport deviations can be considered significant, or not. Section 3.1 of the paper provides to the reader a comprehensive overview on how the specific parameterization of the

MPTRAC model (following the FLEXPART model, Stohl et al., 2005) affects the simulation results. Throughout the revised manuscript we rephrased to "*parameterized* diffusion and subgrid-scale wind fluctuations" to stress that this is referring to model results.

In general I feel the paper attempts to do to much and presents too many results with too much detail (especially figures 7, 9, 10, 12, and 13). The results are comprehensive rather than illustrative, which makes it quite difficult for the reader to synthesize the results and extract the essential points. Perhaps the complete results could go into appendices or supplementary materials (as tables?), with only the most significant results discussed in the main text.

We agree that the results presented in Figs. 7, 9, 10, and 12–14 are rather comprehensive. However, we would prefer to keep them in place for future reference. Following earlier comments, we rearranged several of the plots so that they are more easy to understand. We also moved Sect. 2.2.2 into an appendix of the paper to make the main text more concise.

**Reviewer #2**

This is a well-written, well-designed, and well-structured manuscript. The authors have performed a careful analysis of transport characteristics and their differences between ECMWF's ERA-Interim and ERA-5 reanalyses. I have only a few suggestions and corrections for the authors.

Thank you for this encouraging statement!

Page 9, lines 7-8: suggest revising "should be conserved" to "can be conserved" or "are often mostly conserved". As acknowledged later in the manuscript, conservation of PV and other fields relies on a set of assumed conditions or histories of the tracked air parcels, which are rarely met in the troposphere and often not met in the UTLS.

We rephrased this to "can be conserved" and added another clarifying statement following Reviewer #1.

Page 9, line 29: delete extra "the"

Done.

Page 12, lines 31-32: "...by means of the mean relative..." is poor phrasing. Suggest revising to "...using the mean relative..."

We rephrased this as suggested.

Section 3.4: Suggest adding text to the third and fourth paragraphs outlining when potential temperature and PV are not conserved, as was done for specific humidity here.

We added "Potential temperature and potential vorticity are conserved in reversible adiabatic processes and will not change in the absence of heating, cooling, evaporation, condensation, or mixing (e. g., Curry, 2015; McIntyre, 2015)."

Page 18, lines 2-3: assuming my interpretation is correct, suggest clarifying "(e.g., by means of ensemble simulations)" to "(e.g., by means of ensemble trajectory simulations)". If not, please clarify as appropriate.

We intended to refer to ensembles of trajectory simulations and fixed this as suggested.

Figure 1: for the horizontal axis, I would suggest using "layer depth" instead of "layer width" given the focus on the vertical dimension.

We fixed this as suggested.

Figure 5: The labels for each panel here are confusing. They say "w/o diffusion", but what is being shown are deviations due to diffusion and subgrid-scale mixing. Please revise/simplify these labels and the caption to avoid confusion. This is also true for Figure 6!

We deleted the plot titles in Fig. 5 and simplified the plot titles in Fig. 6 to show only information that was not already provided in the captions.

Figure 6: In addition to the label issue, how are these particles collected for analysis? Based on initial position, final position, some other way? Its not clear how these regional analyses are done. A simple clarification in the caption should suffice.

This question may also apply to the different height ranges and to other figures. We added the following explanation at the end of Sect. 2.3.1: "Here, the binning of the particles into the different height ranges and latitude bands was performed at each time step according to their actual positions along the trajectories."

Figure 12: I suggest labeling each panel by experiment number to improve reader evaluation.

We improved Fig. 12 and rephrased the caption to identify the three downsampling experiments more clearly.

**Reviewer #3**

This work is a very useful comparison of the transport properties of the new ECMWF reanalysis ERA5 compared with the ERA-I and should help in popularizing the usage of ERA5. There are, however a few points that need improvements or clarifications as listed below

Thank you for this encouraging statement!

1) I am unsure that the description of meteorological condition during 2017 is relevant to the scope of this work. This subsection and the two figures 2 and 3 are not exploited in the sequel and are only distracting.

Next to providing a discussion of the meteorological conditions during the year 2017, Section 2.1.2 also illustrates some of the differences between the ERA5 and ERA-Interim data sets. In the revised manuscript, we made sure Figs. 2 and 3 (and the new Fig. 4, see reply to Reviewer #1) are referenced more often and better exploited in the discussion.

2) I am also unsure that the details of disk storage at FZJ in 2.2.2 are of public interest if the ERA5 deposit is only for internal usage.

We agree that the material presented in Sect. 2.2.2 is mostly of a technical nature and may only be interesting for a smaller number of readers. We turned this section into an appendix of the paper, so that the main text becomes more focused and concise.

3) The dispersion of parcels by diffusion is effective only over a couple of days. The subsequent dispersion is due to the explicit wind shear and strain of the resolved eddies. This needs to be clarified with the help of figure 5. Actually, it seems that we see several regimes in figure 5: a diffusive regime with a sqrt(t) behavior (that generates diverging RVTD at small t), an exponential regime characteristic of chaotic dispersion and a linear regime due to large eddy dispersion.

Following comments of Reviewer #1, more details regarding the impact of the different horizontal and vertical components of the parameterization scheme for diffusion and subgrid-scale wind fluctuations have been added to the discussion of Fig. 4. We also revised the discussion of Fig. 5 to point out the different regimes with distinct scaling behavior as referred to by the reviewer.

4) The very good conservation of the potential temperature in the ERA5 compared to ERA-I is an interesting and somewhat puzzling point. It could be due to the improvement in the transport in the model or to the fact that the troncature of the model is more consistent with the spectrum of motion and that rejected modes that generate aliases are only weakly excited. Another possibility is that the temperature assimilation increments are much reduced in ERA5 with respect to ERA-I. In any case, this circumstance should facilitate the determination of appropriate vertical diffusivity to represent the lacking subgrid-scale motion in Lagrangian trajectories. It is quite possible that the required value should be smaller than 0.1 m2 s-1 found in previous studies based on ERA-40 winds.

A more detailed analysis and tuning of the parameterization scheme for diffusion and subgrid-scale wind fluctuations will be a subject of future work. For the present study, the simulation of these effects is based on the given scheme of Stohl et al. (2005). As the differences between ERA5 and ERA-Interim are assessed independently and without simulating diffusion, the diffusion estimates are presented here simply as guiding values, to help judge if other transport deviations can be considered significant, or not. Please see corresponding replies to similar comments provided by Reviewer #1.

5) It is important to mention that convective properties are quite different in the ERA5 which displays much more intermittency than the ERA-Interim.

We added a corresponding statement at the end of Sect 3.2.

6) I do not think that the layer 8-16 km corresponds to the UT/LS in the tropics as it encompasses the mean layer of convective detrainment at 12-13 km.

We added in Sect. 2.3.1: "For the UT/LS region, this definition is particularly limited, as the UT/LS may cover height ranges from roughly 5 to 22 km in reality (Eyring et al., 2010)."

7) I do not see what additional information is brought by figure 11.

This figure serves to illustrate the effect of improved conservation of potential temperature in the Lagrangian transport simulations. Although this effect has been statistically quantified in Fig. 10c,d, the additional Fig. 11 helps to point out this finding more clearly.

8) The comment about assimilation increments of the vertical velocities must be removed or rewritten since there is no assimilation increment of the vertical velocities for hydrostatic models where the vertical velocity is a diagnosed quantity.

We fixed this mistake and clarified that we are referring to assimilation increments in temperature.

9) The downsampling procedure does not really leads to a fair comparison between ERA5 and ERA-I. The spatial downscaling can be seen as a smoothing but the temporal downscaling cannot since the archived wind data are not time integrated quantities but instantaneous values. This section is quite important as users might not all be able to download and store hourly ERA5 data at maximum spatial resolution.

We followed this argument and replaced the downscaling experiment IV by a version in which temporal low-pass filtering has been skipped and only spatial low-pass filtering has been considered. Most transport deviations remained similar, but a notable exception was found for vertical transport deviations in the stratosphere, which we attribute to aliasing effects. The discussion in Sect. 3.5 was revised accordingly.

[revised manuscript text omitted]